# IGHMBP2 deletion suppresses translation and activates the integrated stress response

Jesslyn Park[1,2], Hetvee Desai[1], José M Liboy-Lugo[1,2], Sohyun Gu[1], Ziad Jowhar[1,3], Albert Xu[1,3], Stephen N Floor[1,4]

**IGHMBP2 is a nonessential, superfamily 1 DNA/RNA helicase that is mutated in patients with rare neuromuscular diseases SMARD1 and CMT2S. IGHMBP2 is implicated in translational and transcriptional regulation via biochemical association with ribosomal proteins, pre-rRNA processing factors, and tRNA-related species. To uncover the cellular consequences of perturbing *IGHMBP2*, we generated full and partial IGHMBP2 deletion K562 cell lines. Using polysome profiling and a nascent protein synthesis assay, we found that IGHMBP2 deletion modestly reduces global translation. We performed Ribo-seq and RNA-seq and identified diverse gene expression changes due to IGHMBP2 deletion, including *ATF4* up-regulation. With recent studies showing the integrated stress response (ISR) can contribute to tRNA metabolism-linked neuropathies, we asked whether perturbing *IGHMBP2* promotes ISR activation. We generated ATF4 reporter cell lines and found IGHMBP2 knockout cells demonstrate basal, chronic ISR activation. Our work expands upon the impact of IGHMBP2 in translation and elucidates molecular mechanisms that may link mutant IGHMBP2 to severe clinical phenotypes.**

## Introduction

IGHMBP2 is a ubiquitously expressed, superfamily 1 (SF1) DNA/RNA helicase implicated in the regulation of mRNA translation (Grohmann et al, 2001; de Planell-Saguer et al, 2009). IGHMBP2 loss-of-function mutations cause the rare autosomal recessive diseases spinal muscular atrophy with respiratory distress type 1 (SMARD1) and Charcot-Marie-Tooth disease type 2S (CMT2S), which are characterized by severe neurodegeneration and myopathies (Grohmann et al, 2001; Lim et al, 2012; Cottenie et al, 2014). Although the mechanism of cellular pathogenesis upon functional loss of IGHMBP2 is unknown, IGHMBP2-dependent translational disruption is a favorable hypothesis considering the relevance of translational dysregulation among inherited disease mechanisms (Scheper et al, 2007; Bohnsack et al, 2023). For example, a prevalent cause of

amyotrophic lateral sclerosis are expansions in *C9orf72*, which lead to global translational repression that inhibits the mRNA decay activity of UPF1, another SF1 helicase (Sun et al, 2020).

To-date, the connection between IGHMBP2 and translational regulation has mostly been evidenced by prior biochemical work and using model mice or mouse motoneurons. Localized translation is slowed in IGHMBP2-deficient motoneurons as indicated via FRAP of a translationally regulated GFP reporter (Surrey et al, 2018). IGHMBP2 is predominantly cytoplasmic and co-sediments with ribosomal proteins and subunits and thus interacts with translation machinery during translation (Grohmann et al, 2004; Guenther et al, 2009a). IGHMBP2 also interacts with translation-relevant macromolecules, such as eIF4G2, elongation factors, and tRNA-Tyr (Guenther et al, 2009a; de Planell-Saguer et al, 2009). Mice with spontaneous *IGHMBP2* mutations–which model human SMARD1 phenotypes–are rescued when expressing a genetic modifier locus encoding *ABT1* and several tRNA genes, further implicating a connection between tRNAs and IGHMBP2 in disease pathology (Cox et al, 1998; de Planell-Saguer et al, 2009).

Recent work has established a connection between tRNA and diverse neurodegenerative disorders. Disruption in tRNA regulatory genes via mutations in various aminoacyl-tRNA synthetases (ARS) causes several axonal types of CMT, such as CMT2N, CMT2U, CMT2W, and CMT2D, and can also cause other forms of neurodegeneration or ataxia (Lee et al, 2006). Recently, defective GARS activity in CMT2D model mice was found to induce ribosomal stalling and activate GCN2 kinase, triggering the integrated stress response (ISR), which is a disease mechanism among several other neurodegenerative disorders (Costa-Mattioli & Walter, 2020; Mendonsa et al, 2021; Spaulding et al, 2021). Upon sensing various stimuli such as tRNA deficiency, viral infection, ER stress, and oxidative stress, ISR kinases phosphorylate eIF2α, which up-regulates activating transcription factor 4 (ATF4) and promotes cellular reprogramming toward survival or apoptotic pathways (Costa-Mattioli & Walter, 2020). It is unknown whether the ISR is activated by loss of IGHMBP2.

In this study, we investigated the cellular and molecular consequences of IGHMBP2 deletion in human cells. We found that IGHMBP2 depletion causes cell proliferation and global protein

---

[1]Department of Cell and Tissue Biology, University of California, San Francisco, San Francisco, CA, USA  [2]Tetrad Graduate Program, University of California, San Francisco, San Francisco, CA, USA  [3]Biomedical Sciences Graduate Program, University of California, San Francisco, San Francisco, CA, USA  [4]Helen Diller Family Comprehensive Cancer Center, University of California, San Francisco, San Francisco, CA, USA

Correspondence: stephen@floorlab.org

synthesis to decline, supporting a role of IGHMBP2 in global translational efficiency. We defined the gene expression changes caused by loss of IGHMBP2 using RNA-seq and ribosome profiling and found that expression of the stress-induced transcription factor ATF4 was up-regulated in IGHMBP2 deletion cells. We generated ATF4 reporter cell lines to confirm the occurrence and reversibility of chronic, IGHMBP2-associated activation of the ISR. Our observation of ISR activation and translational suppression caused by IGHMBP2 deletion may point to a possible disease mechanism and solidifies a role for IGHMBP2 in translational regulation.

# Results

## IGHMBP2 deletion impacts cellular proliferation and global translation

To investigate the molecular function of IGHMBP2, we stably perturbed its expression in K562 CRISPRi cells using CRISPR-Cas9 with two sgRNAs targeting exon 2 of *IGHMBP2* (Fig 1A). Edited clones were screened for inactivated alleles via PCR, and selected partial and full-knockout IGHMBP2 monoclonal cell lines were validated by Sanger sequencing and Western blot (Figs 1B and C and S1A–E). To determine the physiological impact of IGHMBP2 deletion in cells, we stably expressed mEGFP in IGHMBP2 deletion clones and measured competitive proliferation relative to parental cells via flow cytometry across 2 wk of cell passages. We observed lower proliferation rates and heterogenous cell morphology among IGHMBP2 deletion cells (Figs 1D and E and S1F). We confirmed mEGFP expression does not impact cell proliferation and that cell lines expressing an alternate fluorophore used in subsequent experiments, TagBFP, exhibited comparable proliferation results (Fig S1G). Expression of fluorescent proteins remained robust across the experimental timeframes (Fig S1H). We conclude that IGHMBP2 loss induces a modest proliferation defect in K562 cells.

The effect of IGHMBP2 on proliferation could be related to its implicated role in protein synthesis. We therefore assessed the impact of IGHMBP2 deletion on global translation by performing polysome profiling (Fig 2A). Western blot of polysome fractions shows IGHMBP2 co-sediments with 40S, 60S, and 80S ribosomal species (Fig 2B). When we quantified the area under the curve for polysome peak groups and compared relative peak ratios per cell line, we observed mild yet reproducible enrichment of free ribosomal subunits relative to monosomes and polysomes in IGHMBP2 deletion clones, while subunit abundances held constant (Fig 2C). This suggests that translational suppression caused by IGHMBP2 deletion is not because of gross ribosomal biogenesis defects. To monitor global translation in cell lines differentially expressing IGHMBP2 at single-cell resolution, we measured relative rates of nascent polypeptide synthesis via an O-propargyl-puromycin (OPP) assay. We found translation was suppressed in IGHMBP2 deletion clones (Figs 2D and S2A–C), consistent with the polysome profiling results. We reproduced these results in cell lines expressing transgenic mEGFP, which were initially sorted to include all GFP+ cells and detected decreased mEGFP expression per cell among

*IGHMBP2*-disrupted clones as an additional orthogonal measure of translational activity (Fig 2E and F).

## Ribosome profiling reveals differentially expressed genes in IGHMBP2 deletion cells

Modest global changes in translation can result from changes to the translation of a subset of mRNAs (Calviello et al, 2021). To determine the gene expression changes in IGHMBP2 deletion cells, we performed ribosome profiling alongside RNA-seq. Ribo-seq and RNA-seq read counts showed high correlation between replicates (Fig S3A–J). We confirmed that Ribo-seq data exhibited 3-nt periodicity (Fig S4A–J), and most of the reads were forward-sense stranded (Fig S5A–J) and mapped to coding exons as expected (Fig S6A–J). We then performed differential expression analysis to identify and interpret the mode of gene regulation of IGHMBP2-dependent differentially expressed genes (DEGs). For each gene among RNA-seq and Ribo-seq results, IGHMBP2-dependent $\log_2$ fold-changes (L2FC) were determined relative to parental expression levels and the Ribo:RNA-seq interaction term L2FC signified translational efficiency (TE). Genes with significant RNA-seq, Ribo-seq, or TE level changes were classified under ΔRNA-seq, ΔRibo-seq, or ΔTE, respectively. If applicable, ΔTE genes were further categorized as translation exclusive if only ribosome occupancy level was changed, or translation buffered if only mRNA abundance was altered (Chothani et al, 2019). To reduce possible artifacts for subsequent analysis, only genes found significant in at least two of the four ΔIGHMBP2 cell lines in any classification were retained as a significant hit or else designated nonsignificant (Table S1). When broadly comparing results per clone, we observed an increased magnitude and number of gene expression changes in full versus partial IGHMBP2 deletion cells (Fig 3A).

We functionally characterized IGHMBP2 KO differential expression results via gene set enrichment analysis (GSEA; Table S2). We observed gene expression attenuation between IGHMBP2-dependent RNA-seq (Fig 3B) and TE results (Fig 3C): translation machinery and mitochondrial gene sets were up-regulated transcriptionally, yet suppressed in TE. Developmental and homeostatic gene sets declined transcriptionally, yet increased in TE. The *peptide biosynthetic process* set—which contains ribosomal machinery, translation initiation, and elongation factors largely overlapping within other *translation* sets—was down-regulated among IGHMBP2 KO TE results via 167 out of 500 possible genes. Also among IGHMBP2 KO TE results, the *regulation of developmental process* set was highly enriched via 306 of 854 possible genes that broadly overlap with the *anatomical structure morphogenesis* gene set (Fig 3C; Table S2). Altered transcriptional programming due to IGHMBP2-dependent TE changes was also suggested by enrichment of the *negative regulation of DNA-templated transcription factor* gene set, with a normalized enrichment score (NES) of 1.6 (*P*.adj = $5.6 \times 10^{-5}$), reflecting 236 of 582 possible genes (Table S2). Upon performing GSEA with ribosomal footprint fold-changes, we observed translational suppression of various homeostatic gene sets appropriately corresponding with the hemopoietic cell type of K562 cells (Fig 3D).

To visualize reproducible RNA-seq level changes, we filtered for ΔRNA-seq genes in both heterozygous or full IGHMBP2 deletion

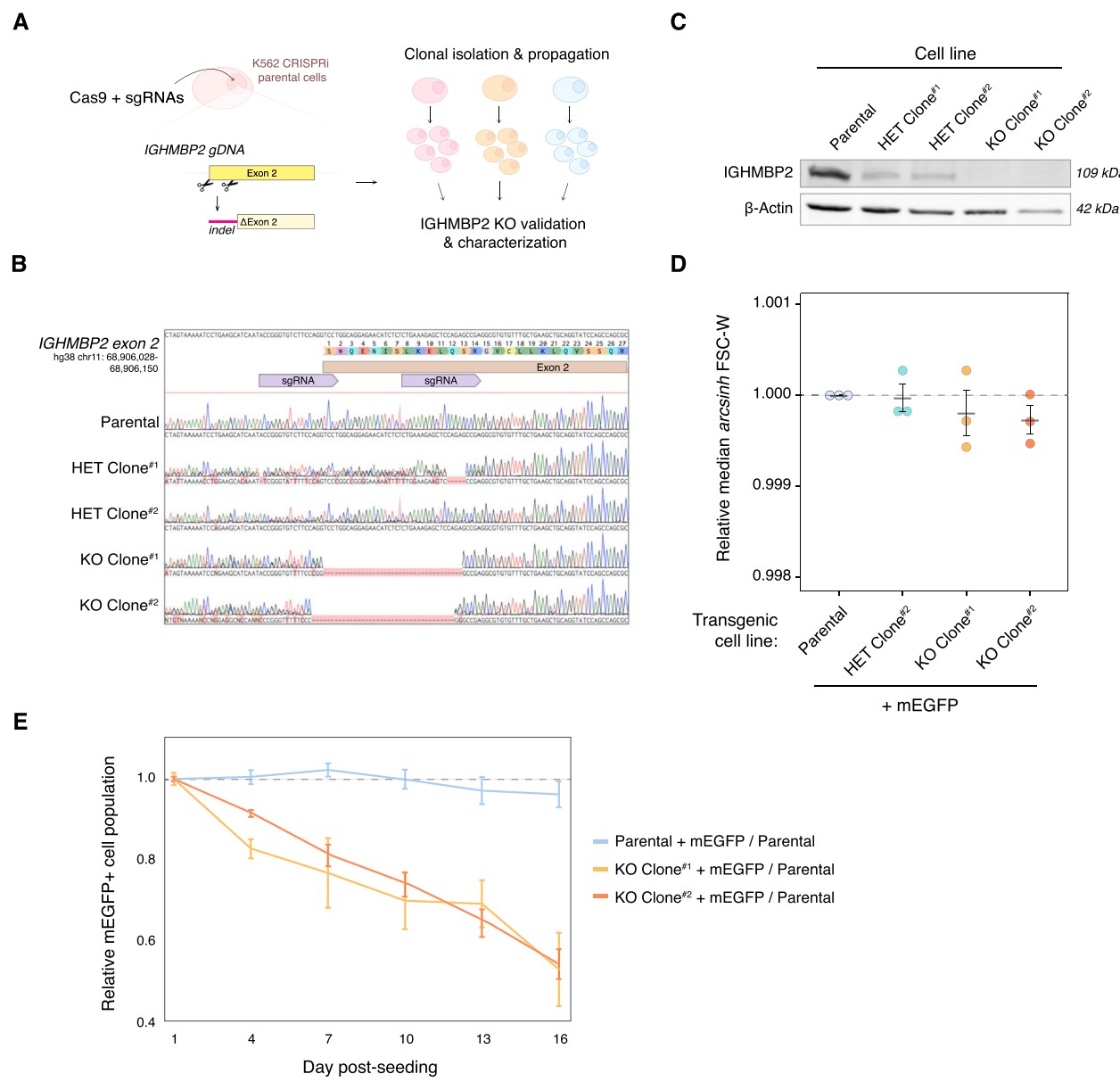

**Figure 1. IGHMBP2 deletion decreases proliferation of cells.**
**(A)** Cas9-mediated IGHMBP2 deletion in K562 CRISPRi cells (parental). **(B)** Sanger sequencing of gDNA surrounding the *IGHMBP2* exon 2 cut-site in clones screened for indels via PCR. Cas9-sgRNA-targeted regions are shown in purple. **(C)** Western blot visualizing IGHMBP2 expression in clones depicted in (B), validating reduced IGHMBP2 protein levels in IGHMBP2 HET Clones #1 and #2 and full deletion in IGHMBP2 KO Clones #1 and #2. **(D)** Median forward-scatter widths (FSC-W) among ΔIGHMBP2 cell lines normalized to the median FSC-W of parental cells per day of measurement for three separate days (n = 3), derived from flow cytometry measurements of at least 2,500 cells per sample 1 d post-passaging. Horizontal black lines show the mean, and error bars show SEM of normalized medians. **(E)** Representative competitive proliferation trends between ΔIGHMBP2 cell lines stably expressing mEGFP seeded with 50% non-fluorescent parental cells. Samples were plated in triplicate wells on Day 0 and independently passaged on each day of measurement. The mEGFP+ fraction per well was normalized to the mean Day 1 fraction per cell line among triplicate wells measured by flow cytometry, and error bars show the coefficient of variation.
Source data are available for this figure.

clones. For each group, most of the hits overlapped in the same directionality (10 out of 13 DEGs between IGHMBP2 HET clones; 43 out of 51 DEGs between IGHMBP2 KO clones) and showed significant correlation (Fig S7A and B). Together, these data identify the genetic changes underlying morphological and translation defects observed as a consequence of IGHMBP2 deletion, again implicating translation as a process impacted by loss of IGHMBP2 (Fig 3E).

## ATF4 is up-regulated in IGHMBP2 deletion cells

Considering the recessive nature of *IGHMBP2*-linked pathologies, we sought to determine the genetic changes specific to full loss of IGHMBP2 to identify possible disease-relevant targets. Of several ΔTE genes identified from our Ribo:RNA-seq analysis (Fig S8A), ATF4 expression was notably up-regulated in genes filtered

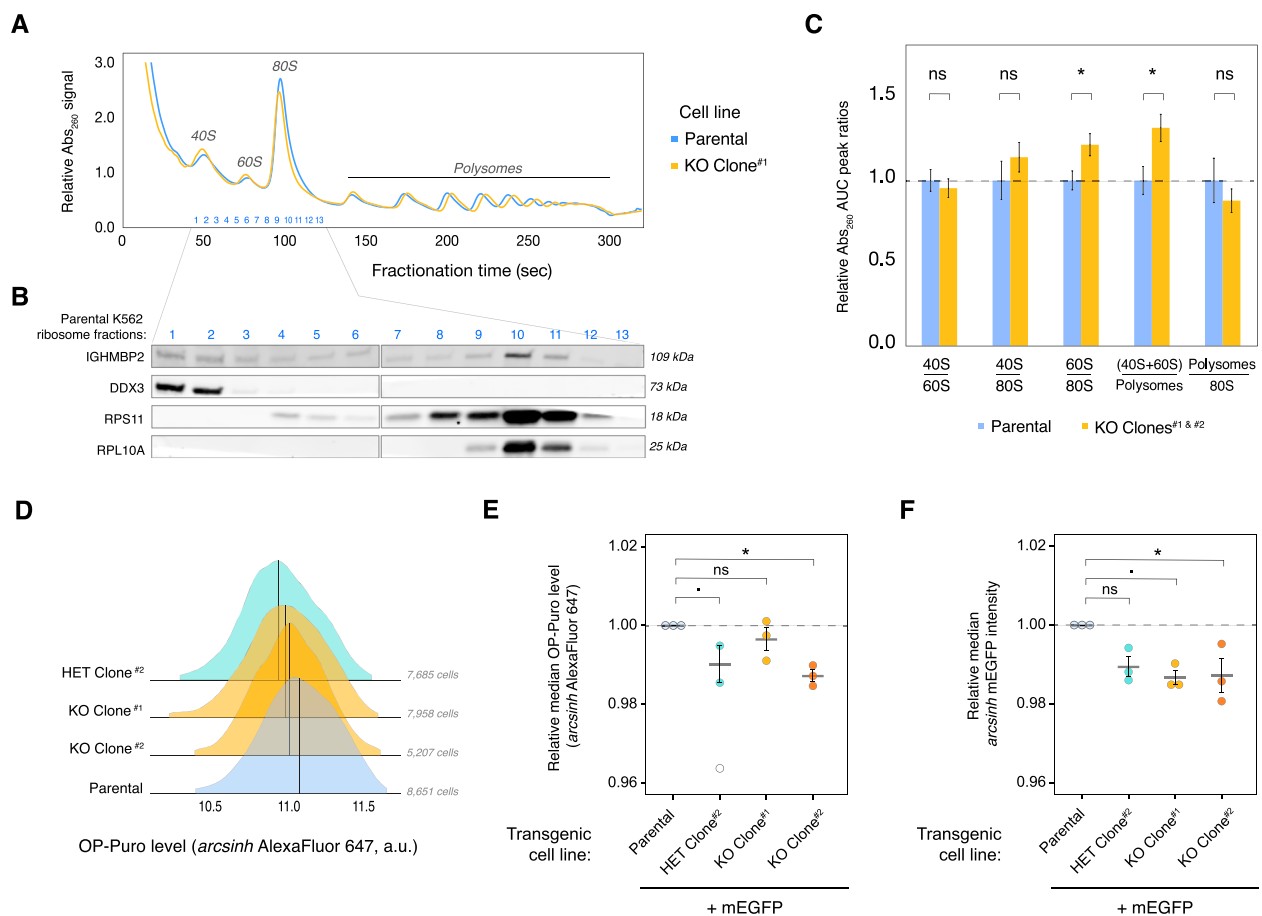

**Figure 2. IGHMBP2 deletion reduces global translation in cells.**
**(A)** Representative polysome profiles from parental K562 (blue) and IGHMBP2 KO Clone #1 (yellow) cell lines. **(A, B)** Western blot of fractions from the parental polysome profile in (A). **(C)** Area under the curve quantification of polysome profiles, for parental (n = 5) and IGHMBP2 deletion clone (n = 7) samples collected from separate flasks, measured on three different days, and normalized to the mean parental signal per day of measurement. Unpaired, two-tailed, two-sample $t$ test was performed between parental versus KO polysome area under the curve ratios (*$P$.adj ≤ 0.05), and error bars show SEM. **(D)** Representative O-propargyl-puromycin levels in ΔIGHMBP2 cell lines quantified via flow cytometry. This experiment was performed three times with comparable results using either Alexa Fluor 647 as shown or Alexa Fluor 594 (Fig S2). Medians are indicated with black lines. **(E)** O-propargyl-puromycin assay results using ΔIGHMBP2 cell lines expressing mEGFP. Values beyond 2.5 SD from the mean of all data points were considered outliers omitted from statistical interpretations (SD = 2.6 for single outlier in HET clone; shown unfilled). **(F)** Relative mEGFP expression in samples from (E). For (E, F), the median fluorescence per cell line is normalized to the median fluorescence of parental cells from the same day, repeated on three separate days (n = 3). Fluorescence of at least 2,300 cells in the final gate per sample was measured by flow cytometry. Horizontal black lines show the mean and error bars show SEM of normalized median values. **(E, F)** One-way ANOVA (Kruskal–Wallis) was performed against all four cell lines, and we rejected the null hypothesis with $P$ = 0.07 for both (E, F). Pairwise statistics were determined via Conover–Iman post hoc testing and shown relative to parental (ns: $P$.adj > 0.1, .: $P$.adj ≤ 0.1, *: $P$.adj ≤ 0.05). Source data are available for this figure.

for dependence on full loss of IGHMBP2 (Fig 4A). Upon assessing ΔTE categorization, *ATF4* was identified as a translation-exclusive ΔTE gene only in the full IGHMBP2 KO genotype condition (Fig S8B and C). Induction of ATF4 is caused by translational control through two upstream ORFs (Lu et al, 2004; Vattem & Wek, 2004). We observed an enrichment of normalized reads mapped to the uORFs and CDS of *ATF4* in IGHMBP2 deletion clones relative to parental cells, especially at the enhancer uORF1 (Fig 4B and C), suggesting translational activation of ATF4 expression.

To test the hypothesis that IGHMBP2 deletion induces ATF4 expression, we generated parental and ΔIGHMBP2 reporter cell lines stably expressing a CMV-driven uORF1,2(ATF4)-mApple reporter gene (Fig 4D). As part of the reporter system, a separate CMV-EGFP transgene was introduced to enable

normalization for promoter strength and translational activity (Guo et al, 2020). We used this reporter because of challenges with robust detection of ATF4 via immunoblot. We detected elevated ATF4 reporter levels in IGHMBP2 deletion clones relative to parental cells (Figs 4E and S9A). We confirmed inducibility of reporter expression upon acute ISR stimulation via thapsigargin (Fig S9A and B). We then exogenously expressed IGHMBP2 N- or C-terminally fused with TagBFP in our IGHMBP2 deletion reporter cell lines and observed ATF4 reporter expression reversed toward parental levels of ATF4 with both transgenes (Figs 4F and S9C and D), indicating genetic rescue and supporting the causative nature of *IGHMBP2* genetic lesions. We therefore find that loss of IGHMBP2 induces ATF4 expression.

**Figure 3. IGHMBP2 loss alters translation of diverse mRNAs.**
**(A)** Ribo-seq versus RNA-seq shrunken log$_2$ fold change (L2FC) per gene from partial or full IGHMBP2 deletion clones relative to parental cells. Differential expression (DE) analysis was performed using Wald test, and *P*-values were adjusted (*P*.adj) via Benjamini–Hochberg method. Cut-offs used for DE classifications are *P*.adj < 0.01 for ΔRNA-seq (pink, green, and orange), and *P*.adj < 0.05 for ΔRibo-seq (green and violet) and Δtranslation efficiency (ΔTE; blue, violet, and orange). ΔTE genes across all clones were identified via likelihood ratio test against the Ribo:RNA-seq interaction term across all samples. Genes of both ΔTE and ΔRibo-seq are identified as translation exclusive (violet). Genes of ΔTE and ΔRNA-seq are classified as translation buffered (orange). The number of differentially expressed genes (n$_{DEG}$) are shown per cell line. **(B, C)** Top 10 up versus down-regulated enriched gene sets using ranked fold-changes from RNA-seq and (C) Ribo:RNA-seq results. **(B, C)** GS labels are orange or pink if expression trends between (B, C) are opposing positively or negatively from RNA to TE levels, respectively. **(D)** Enrichment network map of gene set enrichment analysis results with Ribo-seq L2FCs. In (B, C, D), gene set enrichment analysis was computed using the *Biological Process* ontology; min GS size = 25, max GS size = 1,000 with 100,000 permutations. **(E)** Overview of the cellular impact of *IGHMBP2* disruption.

## IGHMBP2 deletion results in ISR activation

ATF4 is a stress-responsive transcription factor that is tightly regulated at the translational level (Harding et al, 2000; Vattem & Wek, 2004). Given the role of ATF4 as part of the ISR and the

relevance of the ISR in neurodegenerative diseases, we wondered if ISR activation is a consequence of IGHMBP2 deletion. IGHMBP2-dependent, mild suppression of ribosomal proteins and elongation factors shown in our sequencing results would also be consistent as a repercussion of ISR-induced eIF2α phosphorylation (Sidrauski

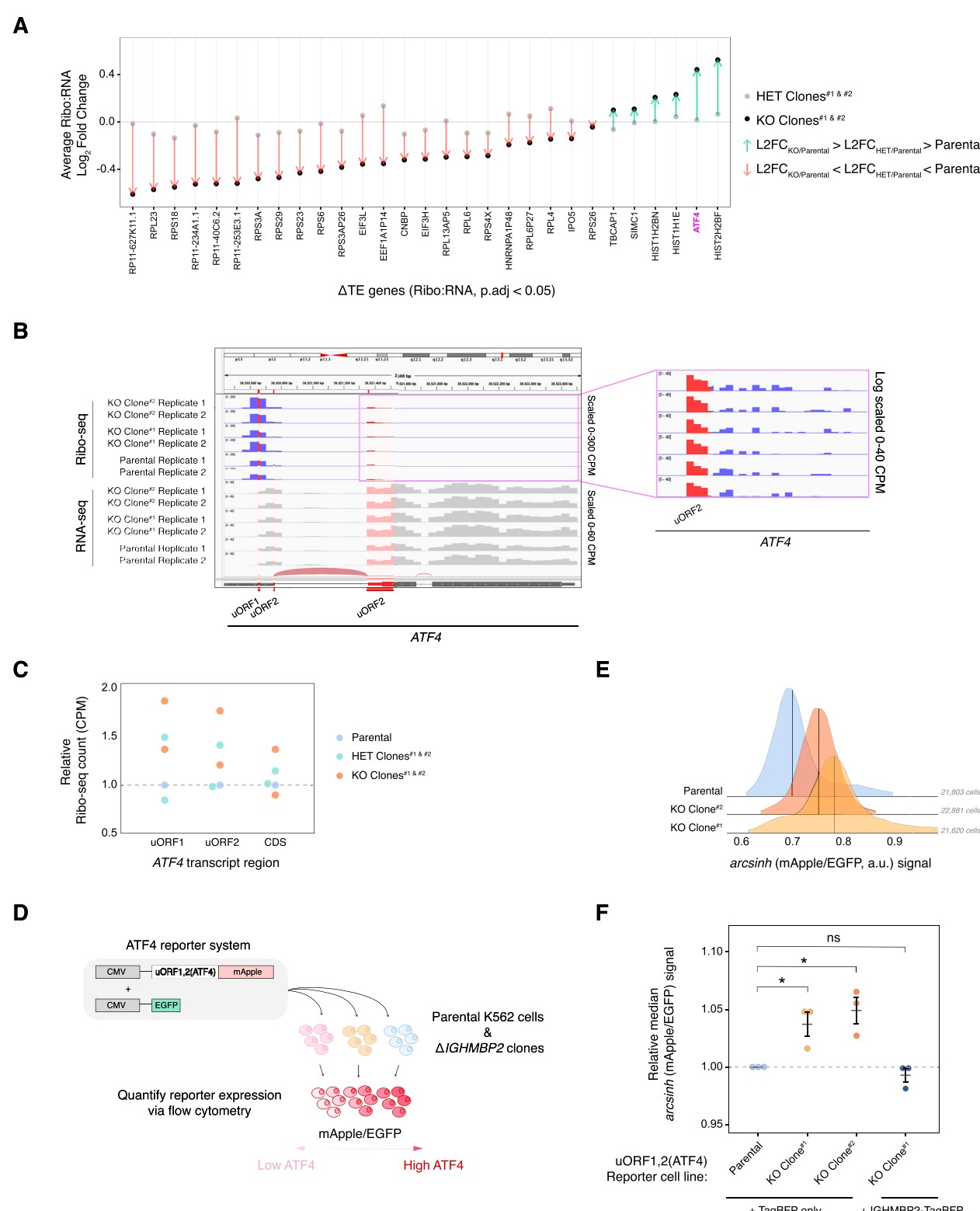

**Figure 4. ATF4 is up-regulated in IGHMBP2 deletion cells.**
**(A)** Average L2FC of a subset of ΔTE genes in full versus heterozygous IGHMBP2 deletion clones relative to expression in parental cells. ΔTE-classified genes were filtered for genes exhibiting average FC within 10% in HET clones and if L2FC$_{Avg}$ was intermediate in IGHMBP2 HET clones relative to L2FC$_{Avg}$ in IGHMBP2 KO clones. **(B)** Ribo-seq reads from IGHMBP2 deletion versus parental samples normalized by counts per million (CPM) and mapped to the uORF regions of *ATF4*. **(C)** Quantification of relative Ribo-seq counts corresponding to *ATF4* transcript regions among ΔIGHMBP2 cell lines. The uORF2 overlap with CDS of *ATF4* was excluded from uORF2 count quantification. Each dot represents the average CPM between two replicates per clone normalized to the parental CPM per indicated *ATF4* region. **(D)** Schematic of ATF4 reporter system.

et al, 2015). Surprisingly, differences in p-eIF2α were not discernible between parental and IGHMBP2 deletion cells via Western blot (Fig S10A and B). This may be due to homeostatic regulation, which produces reduced levels of p-eIF2α upon chronic ISR activation relative to acute stress (Novoa et al, 2003; Guan et al, 2017). However, upon treating IGHMBP2 deletion cell lines with p-eIF2α-dependent ISR inhibitor (ISRIB; Sidrauski et al, 2015), ATF4 reporter expression in IGHMBP2 KO cells robustly reversed toward parental reporter levels at steady state, supporting the occurrence of chronic, p-eIF2α–induced ISR activation (Fig 5A).

Four kinases can activate the ISR: PKR senses specific types of dsRNA, HRI senses heme concentration and mitochondrial dysfunction, PERK responds to the unfolded protein response in the endoplasmic reticulum, and GCN2 is activated via engagement with uncharged tRNAs or detection of stalled ribosomes (Costa-Mattioli & Walter, 2020). Given the genetic connection between IGHMBP2 and tRNA metabolism, we hypothesized ISR activation in IGHMBP2 deletion cells could be driven by empty A-site induced ribosomal stalling associated with tRNA-Tyr availability and consequential GCN2 activation. Considering IGHMBP2-ribosome association biochemically documented in literature and reproduced in our own data (Fig 2B; Guenther et al, 2009a), IGHMBP2 loss could also disrupt translation elongation by some mechanism unrelated to tRNA metabolism, yet again pointing to the GCN2 arm of the ISR. We thus treated ATF4 reporter cell lines with kinase inhibitor GCN2iB (Nakamura et al, 2018) and observed modest effects upon IGHMBP2 KO clones (Fig 5B). Although expected, two-way interaction between ISRIB and GCN2iB-treated parental versus IGHMBP2 deletion cells was statistically insignificant. This may suggest the effect of ISR inhibition on ATF4 reporter expression is not IGHMBP2-dependent, or is driven by the mild effect of ISRIB on parental cells and overall mild effect of GCN2iB on IGHMBP2 KO cells. One-way analysis, however, shows ATF4 up-regulation in IGHMBP2 KO cell lines is partially rescued by GCN2iB treatment, supporting the involvement of GCN2 in ISR activation (Fig 5A and B).

To further test the cellular and molecular consequences of ISR treatment on IGHMBP2 KO cells, we monitored relative growth rates in the presence of ISRIB or GCN2iB and found that proliferation of IGHMBP2 deletion clones was not impacted by either treatments across a 9-d time-course (Fig S11A). However, when measuring relative morphology and translation among cell lines, both inhibitors rescued relative median cell size in IGHMBP2 KO clones after 9 d (Figs 5C and S11B), and ISRIB partially restored translation indicated by transgenic mEGFP expression after 1 wk of treatment (Figs 5D and S11C). The particular strength of ISRIB in translational rescue suggests that targeting p-eIF2α as the ISR bottleneck is more effective than suppressing upstream ISR stimulation via GCN2 inhibition, which may be expected under chronic stress activation. Although GCN2iB treatment does not restore translation in IGHMBP2 deletion cells, which could also implicate involvement of additional ISR kinases, we note its dissimilar effects between parental and heterozygous versus IGHMBP2 KO cell lines. After 5 d of treatment, GCN2iB reproducibly negatively impacts translation only in the presence of IGHMBP2 (Figs 5D and S11C), which may evidence differential IGHMBP2-dependent GCN2 activity or translation buffering effects that cannot be directly measured via net changes with this experiment.

Upon determining that ISR activation is driven by GCN2 in IGHMBP2 deletion cells, we wondered if we could detect ribosome stalling among our Ribo-seq reads, as empty A-site stalled and collided ribosomes activate GCN2 (Ishimura et al, 2016; Inglis et al, 2019). We first performed a metagene analysis using Ribo-seq reads to assess whether IGHMBP2 deletion alters ribosome positioning along transcripts globally, as might occur under strong stalling conditions. We observed consistent ribosomal occupancy across sequenced transcripts in knockout compared with parental cell lines (Fig S12A), which may suggest IGHMBP2 impacts translation by a more ubiquitous mechanism nonspecific to stages of translation, or perhaps toward specific transcripts undetectable by global analysis. Determining whether IGHMBP2-RNA recognition or unwinding activity impacts global translation and how this may be governed by particular biophysical properties or motifs among mRNAs warrants further experimentation.

As tRNA dysregulation may be a favorable hypothesis driving global translation changes, specifically tRNA-Tyr as implicated from prior literature (de Planell-Saguer et al, 2009), we performed a Northern blot to visualize total tRNA-Tyr and tRNA-Val expression among our cell lines (Fig S12B). We did not discern differences in abundance of these tRNAs, suggesting IGHMBP2-dependent differences in tRNA-Tyr levels may not occur in K562 cells or are below detectability with biochemical assays. To further test the hypothesis that IGHMBP2 loss affects decoding of a subset of codons, we applied a recently developed machine learning package, *choros*, to infer in-frame ribosome A-site occupancy of codons globally, but did not detect specific codon enrichment (Mok et al, 2023 *Preprint*; Fig S12C–H). This lack of signal is perhaps explained by the overall subtle effect size of IGHMBP2 loss on translation, or could occur for other unknown reasons. We conclude IGHMBP2 deletion results in chronic, low-level ISR activation mediated by GCN2 signaling. GCN2 recognition of stalled ribosomes (Ishimura et al, 2016) or interaction with uncharged tRNAs (Wek et al, 1995) may occur below our detection threshold in response to disrupted tRNA metabolism or IGHMBP2-ribosome processes as possible ISR activation mechanisms due to IGHMBP2 loss (Fig 5E).

Transcription of lentivirally integrated mApple and EGFP reporters is driven by separate CMV promoters. An integrated stress response-sensitive, synthetic 5′-UTR encoding two uORFs (derived from *ATF4*) is upstream of the mApple ORF. **(E)** uORF1,2(ATF4)-mApple expression normalized to promoter and translational activity (EGFP) in ΔIGHMBP2 reporter cell lines. Medians are indicated with black lines. **(F)** Relative median mApple/EGFP intensities among reporter cell lines expressing TagBFP or IGHMBP2-TagBFP measured by flow cytometry. Measurements were collected on three different days (n = 3), and single-cell mApple/EGFP signals were normalized to the median mApple/EGFP of parental cells from at least 3,000 cells in the final gate per sample. Horizontal black lines show the mean, and error bars reflect SEM of normalized median values. One-way ANOVA (Kruskal–Wallis) was performed against all four cell lines, and we rejected the null hypothesis with $P = 0.03$. Pairwise statistics were determined via Conover–Iman post hoc testing and shown relative to parental (ns: $P.adj > 0.1$, .: $P.adj ≤ 0.1$, *: $P.adj ≤ 0.05$, **: $P.adj ≤ 0.01$).

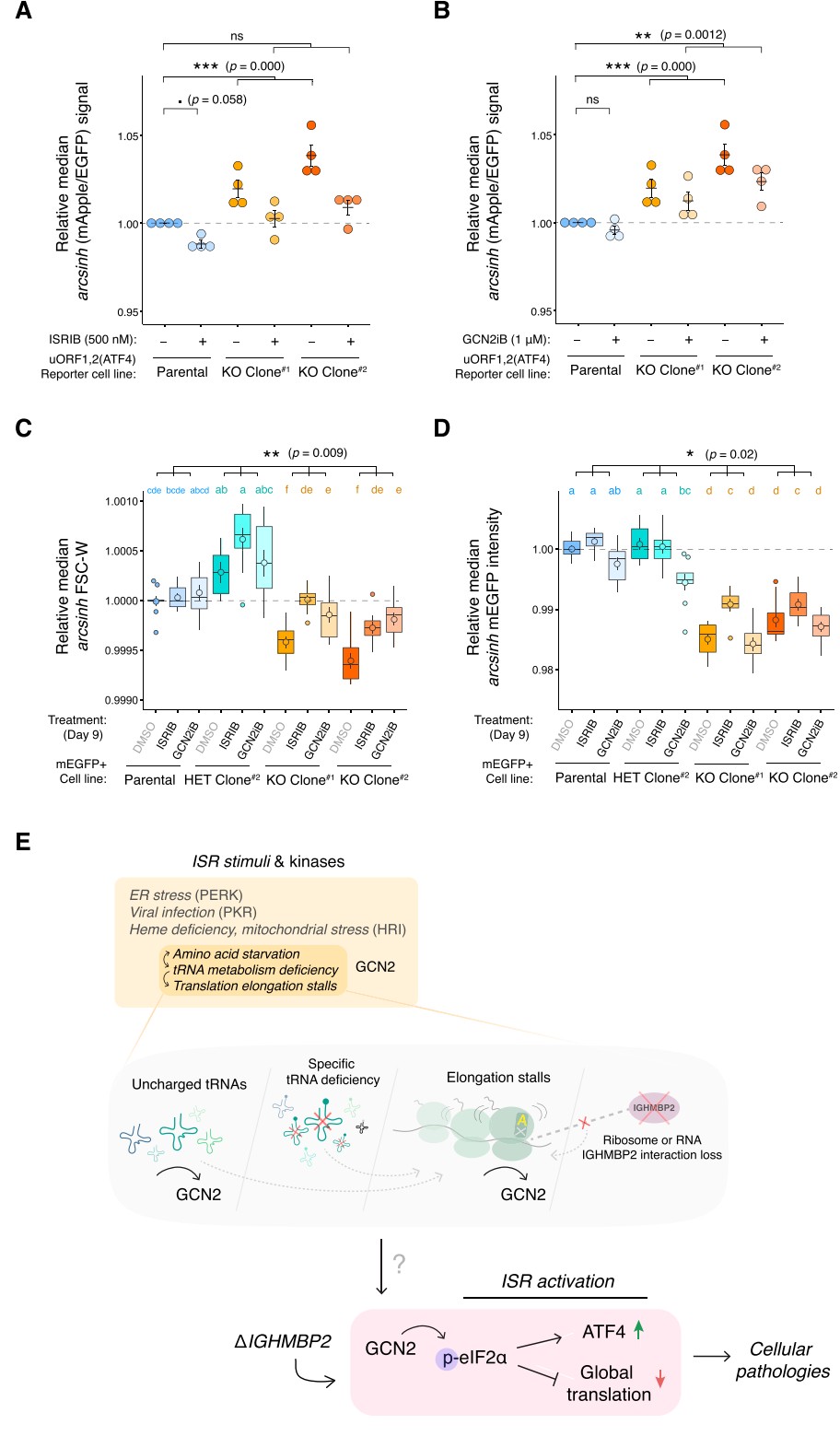

**Figure 5. The integrated stress response is activated in IGHMBP2 deletion cells.**

**(A, B)** Dotplots of relative median mApple/EGFP signal among ΔIGHMBP2 reporter cell lines treated with 500 nM ISRIB or (B) 1 μM GCN2iB for 24 h, measured by flow cytometry. Single-cell mApple/EGFP intensities were normalized to the median mApple/EGFP of untreated (DMSO only) parental cells per day of measurement. For (A, B), normalized medians derived from final gated populations of at least 4,500 cells per sample were determined from experiments performed on four different days (n = 4). Black horizontal lines indicate mean, and error bars show SEM of normalized medians. One-way ANOVA (Kruskal–Wallis) was performed on DMSO, ISRIB, and GCN2iB rank-transformed results under one dataset. The null hypothesis was rejected with $P.\text{adj} = 0$ ($P \ll 0.0001$) and Conover–Iman post hoc test result statistics are shown. **(C, D)** Boxplots of relative median FSC-W and (D) mEGFP fluorescence intensities among mEGFP+ cell lines treated with 500 nM ISRIB or 1 μM GCN2iB for 9 d, normalized to the average Day 1 untreated parental signal from the same plate. Final gated populations of at least 3,000 cells from three independently passaged wells collected across three separate experiments were analyzed (n = 9). Points within boxes indicate mean, and error bars represent SEM of normalized medians. A two-way ANOVA followed by pairwise means comparison via Tukey's post hoc test was performed among rank-transformed parental, heterozygous, and IGHMBP2 KO results to visualize differences between groups. Pairwise compact letter display coloring corresponds to parental (blue), heterozygous (teal), and IGHMBP2 KO (gold) groups for interaction results. The Genotype: Treatment statistics are shown above brackets. For (A, B, C, D), significance codes represent ns: $P > 0.1$, .: $P \leq 0.1$, *: $P \leq 0.05$, **: $P \leq 0.01$, ***: $P \leq 0.001$. **(E)** Possible models for how IGHMBP2 deletion results in GCN2-mediated chronic ISR activation.

## Discussion

In this work, we defined the impact of IGHMBP2 in translation using Cas9-mediated IGHMBP2 knockout human cell lines. We also studied the effects of complete loss of IGHMBP2 expression compared with heterozygous expression to reflect a biologically relevant, non-pathological regime. We show that relative to partial deletion or transgenic knock-in, full loss of IGHMBP2 slows cell proliferation, reduces translation, disrupts gene expression across the transcriptome and translatome, and activates the ISR.

To further characterize ISR activation in IGHMBP2 deletion cells, we performed GSEA with a custom gene set comprising 122 compiled ATF4 target genes (Neill & Masson, 2023). While enrichment of ATF4 target genes in IGHMBP2 KO ΔTE results was similar in magnitude to top gene sets from the *Biological Process* gene ontology, the enrichment score did not reach statistical significance and was null among GSEA with RNA-seq results (Fig S13A–C). ISR-induced apoptotic signaling typically involves CHOP up-regulation (Su & Kilberg, 2008), which we did not detect in our DEG datasets, suggesting cellular stress caused by IGHMBP2 deletion induces a pro-survival ISR program. We notably observed chronic up-regulation of ATF4 at low levels among our IGHMBP2 deletion cell lines, which may also be also associated with pro-survival ISR programming (Pakos-Zebrucka et al, 2016). We surmise pro-survival ISR may enable cells to buffer upstream molecular effects of IGHMBP2 disruption. We were therefore surprised that up to 9 d of ISR suppression via ISRIB or GCN2iB treatment did not measurably affect proliferation rates of IGHMBP2-deficient cells (Fig S11A). We thus speculate that although the ISR alters homeostasis and translation in IGHMBP2 deletion cells–which is supported by morphological and translation rescue upon ISR suppression (Fig 5C and D)–consequences of IGHMBP2 loss orthogonal to ISR stimulation may more directly influence proliferation rate. IGHMBP2 HET cells also exhibited stochastic morphological differences and pronounced decline relative to KO cell growth, yet translation was largely unchanged over time relative to parental cells (Figs 2E and F and S11A–C). This demonstrates how growth processes may be particularly sensitive to deviations in IGHMBP2 abundance, whereas chronically defective translation depends on full loss of IGHMBP2.

Although IGHMBP2 is ubiquitously expressed, IGHMBP2-associated pathogenesis is thought to be tissue-specific. IGHMBP2 is most highly expressed in the brain, and pathological symptoms of SMARD1 and CMT2S are restricted to neural and muscle tissue (Cox et al, 1998; Guenther et al, 2009b; Cottenie et al, 2014). We wonder if this could further explain the small effect sizes in our model in contrast to more measurable ISR effects observed in neuronal cells modeling CMT2D (Spaulding et al, 2021; Zuko et al, 2021). Alternatively, the ISR may be more strongly activated by IGHMBP2 loss in highly proliferative cells such as K562s, thus measuring ISR activation in different cell types or animal models would be important future work. We note that acutely activated ISR promotes restoration of homeostasis through various proposed mechanisms, and it is interesting to postulate how chronic ISR-mediated translational suppression may function to promote survival by attenuating stressors such as eIF2α phosphorylation and ribosomal stalling in an IGHMBP2-deficient context (Pakos-Zebrucka et

al, 2016). Despite the small effect sizes, we robustly and reproducibly observe chronic, reversible translation repression and ISR activation using high-throughput and sensitive approaches such as deep sequencing and flow cytometry. Our work may inform future experimental considerations toward measuring disease-relevant, chronic stress markers that could otherwise remain undetected using traditional techniques.

Whereas tRNA metabolism-driven ISR activation is an intriguing disease mechanism hypothesis, axonal CMT is notably caused by various other sources of molecular dysfunction as well. For example, disruption in mitochondrial activity (CMT2A, CMT2DD, and CMT2EE), cytoskeletal proteins (CMT2B1, CMT2E, and CMT2CC), and other diverse cellular functionalities converge among CMT pathogenesis. Thus, it is feasible that alternative consequences of IGHMBP2 deletion beyond tRNA or elongation dysregulation may more dominantly contribute to neuropathy and myopathy, although it is appealing to speculate this may be tissue-dependent considering SMARD1 model mice are rescued when expressing a modifier locus encoding *ABT1* and five tRNA-Tyr genes (de Planell-Saguer et al, 2009). Most recently, IGHMBP2-dependent differential gene expression has been attributed to translational regulation of the THO complex (THOC) mRNA export proteins in reprogrammed SMARD1 patient-derived fibroblasts and IGHMBP2-knockdown HeLa cells, where mild ribosome stalling was also detected (Prusty et al, 2024). Although THOC expression was unchanged in our K562 ΔIGHMBP2 DEG data, these different results may be due to cell type-specific effects. How mechanisms of IGHMBP2-associated pathogenesis may compound, differentially manifest, or alter in magnitude across different tissue types will be important to deconvolve in future studies.

Consequences of IGHMBP2-DNA impairment remain unaddressed in our study and would be important to distinguish considering IGHMBP2 is also found in the nucleus (Grohmann et al, 2004). The namesake of IGHMBP2 refers to its recognition of DNA motifs that aid immunoglobulin heavy chain recombination, and IGHMBP2 exhibits slower, short-lived dsDNA unwinding processivity in vitro relative to the prototypical SF1 helicase, UPF1 (Mizuta et al, 1993; Kanaan et al, 2018). Expanded functional insight toward IGHMBP2-DNA and its unique processive behavior is yet to be determined. Whether differential transcription caused by IGHMBP2 deletion precedes cellular stress activation and translational dysregulation remains unclear, and this is confounded by transcriptional reprogramming chronically mediated by the ISR. Future DNA-binding experiments may elucidate the possible relevance of IGHMBP2-DNA disruption toward cellular pathology.

One of 95 broadly elusive, nonredundant human helicases, IGHMBP2 is an understudied nonessential gene with recent findings that shed light on its fundamental importance (Umate et al, 2011). IGHMBP2 was predicted to contribute to cellular fitness in a CRISPR screen in HEK293T cells (Hart et al, 2015), and we establish that perturbing IGHMBP2 indeed also reduces cellular fitness in K562 cells via translation suppression and ISR activation. Beyond the physiological consequences we have identified, a recent screen performed in HeLa cells showed IGHMBP2 disruption alters nucleolar dynamics, further demonstrating its ubiquitous impact across non-neuronal cellular models (Sheu-Gruttadauria et al, 2023 *Preprint*). This study also predicted gene functionality based on

how genetic perturbations influenced nucleolar dynamics, which categorized IGHMBP2 in a non-ribosomal biogenesis role that disrupts nucleolar rRNA composition, in-line as a potential consequence of IGHMBP2-associated global translation suppression. Promisingly, adeno-associated virus (AAV) therapeutic strategies have been shown to rescue SMA-related phenotypes, including SMARD1 in mice using AAV9-IGHMBP2 (Shababi et al, 2016; Pattali et al, 2019; Saladini et al, 2020). Thus, our demonstration of reversible IGHMBP2-dependent pathologies may present a therapeutically relevant, cell-based system for further study. Defining the fundamental function of IGHMBP2 and repercussions immediately downstream of IGHMBP2 deletion will also be imperative for identifying alternative, drug-based therapeutics. Our work demonstrates the utility of studying neuropathy-associated gene *IGHMBP2* in an immature human cell line, expands on the fundamental relevance of IGHMBP2, and reinforces the significance of translation in cellular health.

# Materials and Methods

## Cell culture

K562 CRISPRi cell lines used in this study were maintained in RPMI 1640 (*Gibco*) containing L-glutamine and 25 mM HEPES, supplemented with 10% FBS and 1% penicillin-streptomycin. Cells were stored in humidified incubators at 37°C with 5% $CO_2$ and routinely checked for mycoplasma (*Lonza*).

## Cell line generation

Cas9-mediated knockout of IGHMBP2 was performed using two sgRNAs (*Synthego*) to target exon 2 of IGHMBP2 in combination: 5′-CAGAGAGAUGUUCUCCUGCC-3′ and 5′-CUGAAAGAGCUCCAGAGCCG-3′. Ribonucleoprotein (RNP) complexes were prepared by combining 50 pmol of recombinant Cas9 to 50 pmol of each sgRNA and incubating the mixture for 20 min at 37°C. K562 CRISPRi cells (Gilbert et al, 2014) were nucleofected (*SF Cell Line 4D kit; Lonza*) with the RNP. Nucleofected clones were isolated by limiting dilution and indel-containing clonal populations were screened by extracting gDNA using Quick Extract (*Biosearch Technologies*) for PCR amplification using primers targeting up and downstream the Cas9 cut-site: 5′-GGTTGTGGCATTAACTGCCC-3′ and 5′-CCCA-CATCAATTGTTGGAC-3′. PCR products were separated on a 3% agarose gel at 100 V, and *IGHMBP2* gDNA disruption in select clones was further validated via Sanger sequencing with primer 5′-CTTTACGAGGGTACAAGTCACGG-3′ and Western blotting.

## Transgenic cell line generation

For virus generation, Lenti-X HEK 293T cells (*Takara Bio*) were transfected with lentiviral packaging plasmids and transgenic constructs. Virus was harvested and clarified through a 0.45-*µm* filter after 3 d. Lentiviral titering was optimized to result in <50% transduction to generate cells with MOI < 1, and cells were infected via centrifugation at 150*g* for 1 h in 8 *µg*/ml polybrene (*EMD*

*Millipore*). After 3 d, transduced cells were selected via FACS using an 85–100 *µm*-diameter nozzle. ATF4 reporter cell lines were generated and sorted following stress induction as previously described (Guo et al, 2020).

## Western blot

Cell lines were harvested during active growth ($0.4 \times 10^6$–$0.6 \times 10^6$ cells/ml) and lysed in RIPA buffer supplemented with protease inhibitor cocktail (*Halt*). Lysates were centrifuged at 16,000*g* for 20 min at 4°C, and total protein concentration was quantified by Pierce BCA Assay (*Thermo Fisher Scientific*). SDS loading buffer was added to 50 *µg* total protein, and samples were boiled at 95°C for 5 min. Samples were loaded on 4–15% pre-cast PAGE gels (*Bio-Rad*), run at 85 V in running buffer, and transferred to a PVDF membrane (*Bio-Rad*). Membranes were incubated for 1 h in blocking buffer (1% [wt/vol] BSA in 0.1% TBS-T), followed by an overnight incubation at 4°C in blocking buffer containing primary antibody. Membranes were washed for 10 min with 0.1% TBS-T three times, followed by 1 h incubation with secondary antibody with rotation at RT and protected from light. Membranes were washed for 10 min 0.1% TBS-T three times and imaged using a LI-COR Odyssey DLx.

Antibodies used for Western blotting include rabbit anti-IGHMBP2 at 1:500 (23945-1-AP; *Proteintech*), rabbit anti-*β*-actin conjugated to Alexa Fluor 680 (*Abcam*); rabbit anti-eIF2*α* at 1:1,000; rabbit anti-eIF2*α* phospho-S51 at 1:1,000. Membranes were incubated with goat anti-rabbit 800 CW secondary antibody (926-32211; *LI-COR*) at 1:10,000.

## Northern blot

Northern blotting was performed using NorthernMax Kit (*Invitrogen*) with adaptations as described. Total RNAs were extracted from cells with Trizol reagent (15596026; *Invitrogen*). 5 *µg* of total RNA per sample was mixed with 100 mM final concentration of Tris–HCl (pH 9.0) and incubated for 30 min at 37°C to remove amino acids attached to the 3′ end of tRNAs. Samples were then mixed with 2X Formaldehyde Load Dye (8550G; *Invitrogen*) and heat denatured for 3 min at 70°C. RNAs were loaded onto pre-cast 10% TBE-Urea gels (EC68755BOX; Thermo Fisher Scientific) in 1X TBE. RNAs were separated by PAGE, transferred to a positively charged nylon membrane (AM10104; *Invitrogen*), and cross-linked to the membrane using a UV trans-illuminator (Stratalinker 2400 UV Crosslinker) for 2 min at 1,200 × 100 *µJ*. Membranes were pre-hybridized with pre-warmed ULTRAhyb ultrasensitive hybridization buffer (AM8670; *Invitrogen*) at 52°C for 1 h, followed by addition of 10 pmol of each probe. The membranes were hybridized overnight and washed two times with Low Stringency Wash Buffer (AM8673; *Invitrogen*) with rotation at RT for 5 min. Membranes were probed with either tRNA-Tyr, tRNA-Val_{mAC}, or tRNA-Val_{TAC}, stripped, re-probed for tRNA-Gly_{sCC} as an internal control, and imaged using a LI-COR Odyssey DLx.

tRNA-Tyr and tRNA-Val probes were labeled with IRDye800 RD and the tRNA-Gly probe was labeled with IRDye680RD (LI-COR). Probe sequences used include Gly-sCC (5′-TCATTGGCCRGGAATY-GAACCCGGGYCTCCCRCGTGGWAGGCGAGAATTCTACCACTGMACCACC MAYG/iAzideN/C-3′); Tyr-GTA (5′-TCCTTCGAGCCGGASTCGAACCAG

CGACCTAAGGATCTACAGTCCTCCGCTCTACCARCTGAGCTATCGAAG/iAzideN/G-3'); Val-mAC (5'-TGTTTCYGCCYGGTTTCGAACCRGGGACCTTTCGCGTGTKAGGC-GAACGTGATAACCACTACACTACRGAAA/iAzideN/C-3'); Val-TAC (5'-TGGTTCCACT GGGGCTCGAACCCAGGACCTTCTGCGTGTAAAGCAGACGTGATAACCACTACACTATG GAAC/iAzideN/C-3') (Dittmar et al, 2006).

## Flow cytometry

Flow cytometry was performed on BD LSRFortessa instruments. When the high-throughput screening 96-well plate attachment was used, plate parameters were set to a flow rate of 0.5–1.0 $\mu$l/sec, 20 $\mu$l sample volume, 50 $\mu$l mixing volume, 200 $\mu$l/sec mixing speed, 2 mixes, and 200 $\mu$l wash volume. Laser parameters were adjusted to ~400 V for FSC, 250 V for SSC, 260 V for Violet laser detected with 450/50 bandpass filter (for TagBFP), 350 V for Blue detected with 530/30 bandpass filter (for EGFP and Alexa Flour 488), 580 V for YGD detected with 586/615 bandpass filter (for mApple), 520 V for YGC detected with 610/620 bandpass filter (for Alexa Flour 594), and 320 V for Red detected with 670/630 bandpass filter (for Alexa Flour 647). Analysis was performed and visualized using the *flowCore* and *ggcyto* libraries in R (Van et al, 2018). Data values were *arcsinh* transformed and gated to remove debris and doublets. Derivation of standardized and normalized values was performed among data collected on the same day per each experiment. After all gating was applied for each experiment, final populations comprise at least 2,300 to more than 10,000 cells depending on high-throughput screening collection limitations.

Statistical analyses were performed in R. One-way ANOVA (Kruskal–Wallis) was performed for flow cytometry data shown in Figs 2E and F, 4F, and 5A and B. Although $P < 0.1$ is considered weak significance for Fig 2E and F where the ANOVA $P = 0.07$ for both results, we accepted this threshold to reject the null hypothesis ($H_0$) and proceeded with Conover–Iman post hoc testing, as later experiments robustly reproduced these results with strong statistical significance (Fig S11B). Under $P$-adjusted = $P(|T| \geq |t|)$, the post hoc pairwise $H_0$ is rejected if $P$-adj $\leq \alpha$, where $\alpha = 0.05$, and the Benjamini–Hochberg $P$-adjustment method was used. For time-course drug-treated samples, two-way ANOVA testing was performed on rank-transformed data followed by Tukey's post hoc test.

## Nascent protein synthesis assay

The Click-iT Plus OPP Protein Synthesis Assay Kit (C10457; *Thermo Fisher Scientific*) was used as described in a 96-well format with 0.5 × 10$^6$ cells harvested during active growth per reaction. Cells were co-incubated with OPP in DMSO or 125 nM cycloheximide as a negative control for 30 min before fixation with 4% PFA and permeabilization with 0.25% Triton-X 100 in PBS. Cells were resuspended in PBS with 5% (wt/vol) FBS and differential fluorescent signal between cell lines was observed by flow cytometry.

## Cell proliferation competition assay

Cell lines were seeded in 96-well plates at 1.5 × 10$^5$ cells/ml. In wells where mEGFP+ or TagBFP+ transgenic cell lines were seeded with parental cells, 50% abundance of each cell line was targeted. Flow cytometry data were collected 1 d after seeding, showing 35–50% starting populations of fluorescent cells compared with parental cells. Wells were passaged for subsequent measurements every 3 d for 16 d when measuring untreated cells.

For experiments measuring relative growth amid ISR inhibitor treatments, wells were seeded in either 500 nM ISRIB or 1 $\mu$M GCN2iB, measured by flow cytometry 16 h after seeding as Day 1, and measured upon passaging into fresh medium (containing either DMSO, ISRIB, or GCN2iB) every other day for 9 d. Relative viability was determined relative to untreated Day 1 wells, and fluorescence or FSC measurements were normalized relative to untreated parental/parental +mEGFP per each day of measurement.

## Polysome profiling

Cells were grown in separate flasks in triplicate and treated with 100 $\mu$g/ml cycloheximide for 5 min. 10 × 10$^6$ cells were harvested per sample and lysed with a hypotonic lysis buffer (10 mM Hepes, pH 7.9, 1.5 mM MgCl$_2$, 10 mM KCl, 0.5 mM DTT, 1% Triton X-100, and 100 $\mu$g/ml cycloheximide) and trituration with a 26-G needle. Sucrose gradients were prepared with 10% and 50% sucrose in sucrose gradient buffer (100 mM KCl, 20 mM HEPES, pH 7.6, 5 mM MgCl$_2$, 1 mM DTT, and 100 $\mu$g/ml cycloheximide). For each cell line, 100 $\mu$l lysate was layered atop the sucrose gradient, and polysomal species were separated by ultracentrifugation with an SW 41 Ti swinging-bucket rotor (*Beckman Coulter*) at 36,000 RPM for 2 h at 4°C. Polysome profiles were obtained via injection through a spectrophotometer at a flow rate of 2 ml/min, and absorbance was recorded at 260 nm with sensitivity set to 0.1 on the UV/Vis detector. Gradients were eluted by fractionation, and peak fractions were methanol precipitated and resuspended in equal volumes for identification via subsequent immunoblotting.

## Ribosome profiling

Cells were seeded in fresh medium 16 h before harvest. Cell count was measured near 0.3 × 10$^6$ cells/ml per cell line at time of harvest, reflecting active growth conditions. Ribosomal footprint RNAs were obtained as previously described (McGlincy & Ingolia, 2017; Calviello et al, 2021). Cell lysates were treated with RNase I and subjected to size exclusion chromatography using MicroSpin Columns S-400 HR (*Illustra*). Ribosome-protected RNA was then extracted with Trizol, and RNA corresponding to the size of monosomal footprints (26–34 nt) were isolated by size selection via running RNA samples in a 15% polyacrylamide TBE-Urea gel. Footprint fragments were dephosphorylated and ligated to pre-adenylated oligonucleotide linker (NI-816: 5'-/5Phos/NNNNNTAGACAGATCGGAAGAGCACACGTCTGAA/3ddC/-3') with Mth RNA ligase. Unligated linker was removed with RecJ exonuclease. Reverse transcription was performed on linker-ligated RNA with Protoscript II (primer NI-802: 5'-/5Phos/NNAGATCGGAAGAGCGTCGTGTAGGGAAAGAG/iSp18/GTGACTGGAGTTCA GACGTGTGCTC-3'), and samples were treated with 1 M NaOH to hydrolyze remaining RNAs. cDNA was size-selected with a 15% polyacrylamide TBE-Urea gel and circularized with CircLigase II. rRNA was depleted from the sample with a subtraction oligo pool, and cDNA RT was quantified via qPCR. cDNA libraries were amplified with Phusion polymerase (Forward primer NI-798: 5'-AATGATACGGCGACCACCGAGAT

CTACACTCTTTCCCTACACGACGCTC-3′) using unique reverse index primers (reverse primer: 5′-CAAGCAGAAGACGGCATACGAGATJJJJJJGT GACTGGAGTTCAGACGTGTG-3′) and size-selected with a 15% poly-acrylamide TBE-Urea gel. The quality and concentration of amplified libraries was assessed with a Bioanalyzer (*Agilent*) and Qubit (*Invitrogen*); 5 ng/sample were pooled and sequenced with the Illumina HiSeq 4000 (SE65 66x8x8x0) via the sequencing core of the UCSF Center for Advanced Technology.

### RNA-seq

Total RNA was extracted from lysates using Direct-zol RNA Miniprep Kit (*Zymo*) and NEBNext Ultra II Directional RNA Illumina Library Prep Kit (E7760; *NEB*) with NEBNext rRNA Depletion Kit V2 (E7400; *NEB*). RNA-seq samples were prepared with single index primers (ME6609S; *NEB*). RNA integrity and concentration of amplified libraries were assessed with a Bioanalyzer and Qubit; 5 ng/sample was pooled and sequenced with the Illumina HiSeq 4000 (SE65 66x8x8x0) via the sequencing core of the UCSF Center for Advanced Technology.

### Sequencing pre-processing

FASTQ files were converted to FASTA format. Using *cutadapt*, adapter sequences were removed from RNA-seq and Ribo-seq sample reads, and reads were filtered to a minimum length of 22 nt. Reads were collapsed by UMI, which were then removed. Ribosomal RNAs, repeat RNAs, and other noncoding RNAs were aligned against and removed with *repeatmasker* using bowtie2 2.4.1. Index files were generated using STAR 2.7.5a, GRCh38 primary assembly genome.fa file, and a GENCODE v25 .gtf annotation file with −sjdbOverhang set to 64 or 29 for RNA-seq and Ribo-seq indices, respectively. Filtered reads were then mapped to corresponding indices, generating annotated .bam files for analysis.

The number of uniquely mapped RNA-seq reads (million) for two replicates per cell line were 22.9 and 34.7 for parental; 29.3 and 30.6 for IGHMBP2 HET Clone #1; 26.8 and 29.8 for IGHMBP2 HET Clone #2; 28.3 and 29.8 for IGHMBP2 KO Clone #1; 29.8 and 33.6 for IGHMBP2 KO Clone #2. The number of uniquely mapped Ribo-seq reads (million) for two replicates per cell line were 2.0 and 3.3 for parental; 2.5 and 5.5 for IGHMBP2 HET Clone #1; 3.5 and 3.8 for IGHMBP2 HET Clone #2; 1.2 and 2.6 for IGHMBP2 KO Clone #1; 1.9 and 3.5 for IGHMBP2 KO Clone #2.

### Sequencing analysis

RNA-seq and Ribo-seq results were analyzed using *DESeq2* library in R (Love et al, 2014). Transcript biotypes from the corresponding GTF annotation (GENCODE v25) were matched to all genes, and "rRNA" and "Mt_rRNA"-classified genes remaining after pre-processing were removed. Mapped reads were pre-filtered by performing DESeq analysis on all samples to determine the minimal filterThreshold. The minimal filterThreshold was determined to be 3.7 for RNA-seq samples and 5.8 for Ribo-seq samples, and thus genes with counts below these cut-offs were removed from corresponding datasets. DESeq was run with pre-filtered genes with independentFiltering omitted, as this widened ability to statistically contextualize DE changes among genes otherwise abundantly

filtered out with high independent filterThresholds in nuanced perturbation conditions, especially among heterozygous clones. Dispersion results were then shrunken using the *apeglm* method.

Because we compare more than two groups (parental, KO, and HET genotypes) a likelihood ratio test was used rather than default Wald testing, which uses standard error. The initial design used was ~ SeqType + Condition + SeqType:Condition, and DESeq was run using likelihood ratio testing with a reduced design ~SeqType + Condition to derive DE results reflecting SeqType: Condition (Ribo:RNA with respect to genotype). DESeq was run with filterThreshold pre-filtered genes with independentFiltering omitted. Because of the modeling design, resultant *P*-values are the same between IGHMBP2 KO or HET clones, and L2FC were independently determined among samples. Results were then filtered by baseMean cut-offs of > 60 for RNA-seq and baseMean > 40 for Ribo-seq. Genes were classified as significant for RNA-seq *P*-adjusted values < 0.01 and Ribo-seq *P*-adjusted values < 0.05. To further account for possible clonal artifacts and outliers, final significant DE genes among RNA-seq and Ribo-seq results must be significant in at least 2 out of 4 unique clones; genes found significant in only 1 out of the 4 clonal cell lines were filtered from the final datasets.

R packages used to assess quality control of sequencing data include *MultiQC*, *Ribo-seQC*, and *RibosomeprofilingQC* (Ewels et al, 2016; Calviello et al, 2019 *Preprint*). To determine localized read mapping coverage for specific genes, such as *ATF4* uORFs and CDS, bamCoverage from *deeptools* (Ramírez et al, 2016) was used to convert RNA-seq and Ribo-seq pre-processed *.bam* files to *.bigwig* and *.bed* files, where reads were normalized to counts per million within bins separated by 50 nt. Counts per million-scaled reads were then mapped to hg38 using Interactive Genomics Viewer. Normalized counts were extracted from *.bed* files per genomic regions analyzed corresponding to *chr22*: 39,520,650-39,520,700 (uORF1); 39,520,750-39,520,800 and 39,521,350-39,521,400 (uORF2, separated by splice-site); 39,521,450-39,522,600 (CDS).

### Gene set enrichment analysis

Using RNA-seq, Ribo-seq, and Ribo:RNA-seq results output from DESeq analysis, the L2FCs per gene were averaged between cell lines per analysis type. Mean L2FCs were then sorted as ranked lists, and GSEA was performed using *fgsea* and *clusterprofiler* R packages using the *Biological Process* sub-ontology (Yu et al, 2012; Korotkevich et al, 2016 *Preprint*; Wu et al, 2021). The minimum and maximum gene set sizes were set to 25 and 1,000, respectively. GSEA was performed with 100,000 permutations, and exported gene sets reflect a *P*-value cutoff of 0.01. The enrichment map plot was generated using the *enrichplot* library.

### Metagene analysis

Metagene analysis was performed using *ribosomeProfilingQC* library in R with *.bam* files and the *coverageDepth* function. The GENCODE v25 .gtf annotation file was used to define UTR and CDS regions. Metagene profiles were then Min-Max normalized between 0 and 1 for comparability between each clone.

## Codon enrichment analysis

Ribo-seq reads were mapped to the transcriptome using STAR 2.7.5a –quantMode TranscriptomeSAM. Transcriptomic *.bam* files were assigned posterior probability values using RSEM (rsem-calculate-expression) with –fragment-length-mean 29 for analysis with the *choros* R library (Mok et al, 2023 *Preprint*). For parental and IGHMBP2 KO Clone #2 samples, A-site offset rule values were determined to be 15, 14, and 16 for frame 0, 1, and 2 for reads between lengths 29–30 nt; 15, 14, and NA for frame 0, 1, and 2 for 28 nt reads; NA, 14, and NA for frame 0, 1, and 2 for 27 nt reads, respectively. A-site offset rules were less discernible in IGHMBP2 KO Clone #1, and thus was omitted from codon enrichment analysis.

# Data Availability

Sequencing data are available at GEO accession number GSE248890.

# Supplementary Information

# Acknowledgements

Sequencing was performed by the UCSF CAT, supported by UCSF PBBR, RRP IMIA, and NIH 1S10OD028511-01 grants. Flow cytometry and cell sorting was performed at the UCSF Parnassus Flow Cytometry CoLab. K-562 CRISPRi cells were gifted by the Jonathan Weissman lab. We thank Matthew Taliaferro and Wilfried Rossoll for crucial conversations during the formation of this project. The authors also thank the Floor lab members for support and helpful feedback, especially Jess Sheu-Gruttadauria, Yizhu Lin, and Sam Kwok for technical advice. This work was supported by the National Institutes of Health R35GM149255 (to SN Floor). SN Floor is a Pew Scholar in the Biomedical Sciences, supported by The Pew Charitable Trusts.

## Author Contributions

J Park: conceptualization, resources, data curation, software, formal analysis, supervision, validation, investigation, visualization, methodology, and writing—original draft, review, and editing.

H Desai: resources, investigation, methodology, and writing—review and editing.

JM Liboy-Lugo: resources, investigation, methodology, and writing—review and editing.

S Gu: resources, investigation, methodology, and writing—review and editing.

Z Jowhar: resources, investigation, methodology, and writing—review and editing.

A Xu: resources, investigation, methodology, and writing—review and editing.

SN Floor: conceptualization, data curation, software, supervision, funding acquisition, investigation, visualization, project administration, and writing—original draft, review, and editing.

## Conflict of Interest Statement

The authors declare that they have no conflict of interest.

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
