## [Reviewer comments · Life Science Alliance]

Life Science Alliance

IGHMBP2 deletion suppresses translation and activates the integrated stress response

Jesslyn Park, Hetvee Desai, Jose Liboy-Lugo, Sohyun Gu, Ziad Jowhar, Albert Xu, and Stephen Floor

DOI: <https://doi.org/10.26508/lsa.202302554>

Corresponding author(s): Stephen Floor, University of California, San Francisco

Review Timeline:

Submission Date:	2023-12-22
Editorial Decision:	2024-02-05
Revision Received:	2024-04-21
Editorial Decision:	2024-04-26
Revision Received:	2024-05-01
Accepted:	2024-05-02

Transaction Report:

February 5, 2024

Re: Life Science Alliance manuscript #LSA-2023-02554-T

Stephen N Floor
University of California, San Francisco

Dear Dr. Floor,

Thank you for submitting your manuscript entitled "IGHMBP2 deletion suppresses translation and activates the integrated stress response" to Life Science Alliance. The manuscript was assessed by expert reviewers, whose comments are appended to this letter. We invite you to submit a revised manuscript addressing the Reviewer comments.

Thank you for this interesting contribution to Life Science Alliance. We are looking forward to receiving your revised manuscript.

Sincerely,

B. MANUSCRIPT ORGANIZATION AND FORMATTING:

Reviewer #1 (Comments to the Authors (Required)):

Mutations in the gene encoding the DNA/RNA helicase IGHMBP2 are associated with neurological diseases. This manuscript seeks to ask what are the physiological consequences of IGHMBP2 deletion in human cells? Using cultured K562 cells deleted for IGHMBP2 by CRISPR, along with Ribo-seq, RNA-seq, and biochemical and molecular approaches, the manuscript suggests that loss of IGHMBP2 triggers activation of the eIF2 kinase GCN2 and the integrated stress response (ISR). The induction of the IGHMBP2 is suggested to contribute to observed translational changes. Prior studies have shown roles for GCN2/ISR in neuropathologies, and activation of GCN2 was suggested to occur by ribosome stalling/collisions, although other mechanisms are reported for GCN2, such as those involving direct GCN2 activation by uncharged tRNAs. Overall, the manuscript is significant and of interest. The manuscript is clearly written and the conclusions are supported by experimental results. There are a few concerns listed below. Addressing these concerns would bolster the manuscript clarity, rigor, and conclusions.

Reviewer concerns:

1. The introduction is clear, but arguably a bit lengthy. It could be abbreviated some without loss of clarity. With the introduction of the ISR, there should be a sentence to introduce key mechanistic features and ATF4, which is later referred to in the introduction and the manuscript.
2. To enhance readability, some of the supplemental figures labels should be enlarged. For example, the bar graphs in Fig. S5 are wide but the labels are quite small font. Double check the figure legends to be sure that all error bars are defined. Some immunoblot panels are cropped too closely (e.g. Fig. S9) and all should include MW markers.
3. Figure 2A- The western blot panels seem to indicate that the free subunits do not have appreciable ribosomal proteins. How were the lanes normalized?
4. GCN2 activation with modest translational control is documented in the literature. Typically one measures phosphorylation of eIF2 α and GCN2 (along with ATF4 protein), but the ISRIB and GCN2iB results in this manuscript would arguably be sufficient. The manuscript emphasizes ribosome collisions as the likely activator of GCN2 with loss of IGHMBP2. This is understandable given the literature supporting this idea. However, the Ribo-seq data does not appear to support the stalling model, with caveats noted in the manuscript. Other models for GCN2 are strongly suggested in the literature, and uncharged tRNAs (cursor analysis in the manuscript) or other processes should be left as possibilities (last sentence Results).
5. It was not clear whether the combination of GCN2iB and IGHMBP2 depletion has a viability effect in the cultured cells. This would be anticipated, possibly in either direction with loss of GCN2 activity. The ramification of IGHMBP2 depletion activation of the GCN2/ISR is the weakest portion of the manuscript.

Reviewer #2 (Comments to the Authors (Required)):

This manuscript describes the translational effects of deleting IGHMBP2, a gene linked with rare human diseases. Homozygous knockout cells grow slowly and show a modest reduction in overall protein synthesis. Specific mRNAs show larger translational changes, including genes encoding the translational machinery. Transcript-specific analysis also highlights chronic induction of ATF4, the transcription factor responsible for the ISR. This effect is substantially reversed when the ISR kinase Gcn2 is inhibited, although phosphorylation of its target, eIF2 α , is not detectably changed. Likewise, defects are not apparent in either tRNA levels or in altered ribosome profiles.

Taken together, these results strengthen the link between IGHMBP2 and translation, and point towards more specific hypotheses to explain the specific disease phenotypes caused by loss of IGHMBP2. The underlying data are clear, and support the conclusions even when the magnitude of effects is small. I think the readership of LSA will appreciate the contributions of this work and would support its publication, provided the concerns below can be addressed.

1. The CellTiter-Glo assay in Figure S2A, appears to show growth defects in one heterozygous knockout (118) and one homozygous knockout (115), with much less effect in another heterozygous knockout (33) and another homozygous knockout (61). This seems to differ from the other proliferation assays, and these specific results are not discussed in the paper. Is there an explanation for this difference?

2. If the flow cytometry data in 2D is plotted on a logarithmic scale, this should be made clear. And, if these data are indeed log-scaled, are the relative levels in 2E and 2F normalized as linear intensities? If these are relative values on a logarithmic scale (as in, the ratio of two log-scaled measurements) then some justification should be presented for this calculation.
3. Figure 3A shows a number of genes with large (mostly RNA level) changes. How much do the affected genes overlap between different clones? Looking at the data tables, it would seem that there is substantial overlap.
4. In Figure 4C, it would be helpful to display ATF4 CDS occupancy alongside the two uORFs. Also, because uORF2 overlaps the CDS, it would be helpful to specify whether the uORF 2 occupancy measurements included or excluded this overlapping region.
5. In Figure 5A and 5C, are all tests of significance made relative to "Parental" no-drug? Bars or brackets would be helpful to illustrate what comparison was performed. A two-way ANOVA might provide a better statistical analysis here.
6. It is noted that "ribosomal machinery, translation initiation, and translation elongation factors" show significant translational change. This pattern is suggestive of changes in 5' TOP mRNA translation downstream of reduced mTOR activity, has this been considered?
7. It is proposed that eIF2 α phosphorylation is not elevated because of homeostatic regulation. Could effective homeostatic regulation also explain why codon-level effects are not seen in ribosome profiling—that is, translation is reduced to the extent needed to resolve these stalls?

Minor points:

8. Some of the points on Figure 4C seem to overlap, is it possible to offset or jitter them slightly?
9. In Figure S3, should the legend in F read "Ribo-seq" rather than RNA-seq? This would better match the figure title, and explain how A-E differ from F-J
10. In the methods, it is written "tRNA-Tyr, tRNA-Val, or tRNA-Val". This would be less confusing if the anticodons were specified, so it was clear that two different isoacceptors were measured, rather than the repetition reflecting a typographical error.

Reviewer #1

Mutations in the gene encoding the DNA/RNA helicase IGHMBP2 are associated with neurological diseases. This manuscript seeks to ask what are the physiological consequences of IGHMBP2 deletion in human cells? Using cultured K562 cells deleted for IGHMBP2 by CRISPR, along with Ribo-seq, RNA-seq, and biochemical and molecular approaches, the manuscript suggests that loss of IGHMBP2 triggers activation of the eIF2 kinase GCN2 and the integrated stress response (ISR). The induction of the IGHMBP2 is suggested to contribute to observed translational changes. Prior studies have shown roles for GCN2/ISR in neuropathologies, and activation of GCN2 was suggested to occur by ribosome stalling/collisions, although other mechanisms are reported for GCN2, such as those involving direct GCN2 activation by uncharged tRNAs.

Overall, the manuscript is significant and of interest. The manuscript is clearly written and the conclusions are supported by experimental results. There are a few concerns listed below. Addressing these concerns would bolster the manuscript clarity, rigor, and conclusions.

We sincerely thank Reviewer #1 for their positive assessment of our work and helpful feedback.

Reviewer concerns:

1a. The introduction is clear, but arguably a bit lengthy. It could be abbreviated some without loss of clarity.

We appreciate Reviewer #1's suggestion that the introduction could be more succinct. We implemented the following changes, which have reduced word count from to 639 to 543.

Sentence removals:

- *“The disease-associated nonsense, frameshift, or missense mutations in IGHMBP2 are predicted to impair DNA/RNA binding, unwinding, and ATP hydrolysis, and thus pathogenicity is attributed to loss-of-function of IGHMBP2 enzymatic activity (Guenther et al. 2009a; Lim et al. 2012; Saladini et al. 2020).”*
This explanation attributed enzymatic loss-of-function as disease-relevant, but for brevity we removed this statement and specify “loss-of-function” in the preceding sentence.
- *“Defective helicases cause numerous translation-linked pathologies beyond neurodegeneration—DDX3X is a primary example of an SF2 helicase associated with neurodevelopmental disorders and cancers when its translational activities are perturbed (Bohnsack et al. 2023; Gadek et al. 2023).”*
This secondary example is less relevant, as DDX3X is of a different helicase superfamily, and neurodevelopmental disorders are pathologically distinct from neurodegeneration.
- *“Therefore, despite IGHMBP2 having in vitro activity on DNA or RNA, its physiological substrate may be primarily RNA.”*
This logical reasoning may otherwise be implied and is unnecessary given other IGHMBP2-RNA associations exemplified in the introduction.

We also abbreviated and consolidated sentences in paragraph 2 of the introduction to refine unnecessary experimental details.

1b. *With the introduction of the ISR, there should be a sentence to introduce key mechanistic features and ATF4, which is later referred to in the introduction and the manuscript.*

We agree with Reviewer #1 that defining the ISR and the role of ATF4 in the introduction is important. We have addressed this point in our revised paragraph 3 of the introduction.

“... the integrated stress response (ISR), which is a disease mechanism among several other neurodegenerative disorders (Spaulding et al. 2021; Mendonsa et al. 2021; Costa-Mattioli & Walter, 2020). Upon sensing various stimuli such as tRNA deficiency, viral infection, ER stress, and oxidative stress, ISR kinases phosphorylate eIF2 α , which upregulates activating transcription factor 4 (ATF4) and promotes cellular reprogramming toward survival or apoptotic pathways (Costa-Mattioli & Walter, 2020).”

2a. *To enhance readability, some of the supplemental figures labels should be enlarged. For example, the bar graphs in Fig. S5 are wide but the labels are quite small font. Double check the figure legends to be sure that all error bars are defined.*

We thank Reviewer #1 for noting labeling readability; we adjusted Supplemental panel labels for Figs. S1, S2, S4-6, S9, S12-S13. We also greatly appreciate Reviewer #1's attention to detail and have added error bar definitions for Figs. 1D, 4F, and S1G-H.

2b. *Some immunoblot panels are cropped too closely (e.g. Fig. S9) and all should include MW markers.*

We agree this adjustment would enhance our display of blotting data and also have added a Western blot raw images file to our supplementary materials to better address Reviewer #1's feedback. In the updated Fig. S9 (now Fig. S10), we have widened cropping and included expected molecular weights confirmed by corresponding MW markers from protein ladders run in the full blots.

3. *Figure 2A- The western blot panels seem to indicate that the free subunits do not have appreciable ribosomal proteins. How were the lanes normalized?*

We thank Reviewer #1 for their inquiry on sample preparation. We concentrated peak fraction samples, methanol-precipitated protein and resuspended samples in consistent volumes of loading buffer prior to loading to reflect relative concentrations of components corresponding to the polysome profile fractions. We have updated the methods to reflect this with the following statement:

“Gradients were eluted by fractionation and peak fractions were methanol precipitated and resuspended in equal volumes for identification via subsequent immunoblotting.”

Thus, lack of signal for ribosomal proteins may be due to low protein concentration beneath detection thresholds.

4. GCN2 activation with modest translational control is documented in the literature. Typically one measures phosphorylation of eIF2alpha and GCN2 (along with ATF4 protein), but the ISRIB and GCN2iB results in this manuscript would arguably be sufficient. The manuscript emphasizes ribosome collisions as the likely activator of GCN2 with loss of IGHMBP2. This is understandable given the literature supporting this idea. However, the Ribo-seq data does not appear to support the stalling model, with caveats noted in the manuscript. Other models for GCN2 are strongly suggested in the literature, and uncharged tRNAs (cursory analysis in the manuscript) or other processes should be left as possibilities (last sentence Results).

We thank Reviewer #1 for highlighting the need to expand on possible GCN2 mechanisms in our proposed model amid our current experimental scope. We have added additional context surrounding the GCN2 hypothesis and re-worded the last sentence of the results as recommended:

“Given the genetic connection between IGHMBP2 and tRNA metabolism, we hypothesized ISR activation in IGHMBP2 deletion cells could be driven by empty A-site induced ribosomal stalling associated with tRNA-Tyr availability and consequential GCN2 activation. Considering IGHMBP2-ribosome association biochemically documented in literature and reproduced in our own data (Guenther et al. 2009; Fig. 2B), IGHMBP2 loss could also disrupt translation elongation by some mechanism unrelated to tRNA metabolism, yet again pointing to the GCN2 arm of the ISR. [...]”

We conclude IGHMBP2 deletion results in chronic, low-level ISR activation mediated by GCN2 signaling. GCN2 recognition of stalled ribosomes (Ishimura et al. 2016) or interaction with uncharged tRNAs (Wek et al. 1995) may occur below our detection threshold in response to disrupted IGHMBP2-ribosome processes or tRNA metabolism disruptions as possible ISR activation mechanisms due to IGHMBP2 loss (Fig. 5E).”

We also updated our final model in Fig. 5E to include such alternative mechanisms.

5. It was not clear whether the combination of GCN2iB and IGHMBP2 depletion has a viability effect in the cultured cells. This would be anticipated, possibly in either direction with loss of GCN2 activity. The ramification of IGHMBP2 depletion activation of the GCN2/ISR is the weakest portion of the manuscript.

We appreciate Reviewer #1 for raising interest in further experimental testing of the impact of ISR treatment in IGHMBP2 deletion cells and agree that interrogation of associated phenotypic consequences is a very appropriate follow-up to strengthen our overall results. In response, we have performed growth competition experiments (as in Fig. 1E) three times in the presence of ISRIB or GCN2iB treatment and analyzed morphology and translation signals among samples (as in Fig. 1D and 2F). We have revised our manuscript with additional panels in Fig. 5 corresponding to all data shown in a new Fig. S11 and our final results section:

“To further test the cellular and molecular consequences of ISR treatment on IGHMBP2 KO cells, we monitored relative growth rates in the presence of ISRIB or GCN2iB and found that proliferation of IGHMBP2 deletion clones was not impacted by either treatments across a 9-day time-course (Supplemental Fig. S11A). However, when measuring relative morphology and translation among cell lines, both inhibitors rescued relative median cell size in IGHMBP2 KO clones after 9 days (Fig. 5C; Supplemental Fig.

S11B), and ISRIB partially restored translation indicated by transgenic mEGFP expression after 1 week of treatment (Fig. 5D; Supplemental Fig. S11C). The particular strength of ISRIB in translational rescue suggests that targeting p-eIF2 α as the ISR bottleneck is more effective than suppressing upstream ISR stimulation via GCN2 inhibition, which may be expected under chronic stress activation. While GCN2iB treatment does not restore translation in IGHMBP2 deletion cells, which could also implicate involvement of additional ISR kinases, we note its dissimilar effects between parental and heterozygous versus IGHMBP2 KO cell lines. After 5 days of treatment, GCN2iB reproducibly negatively impacts translation only in the presence of IGHMBP2 (Fig. 5D; Supplemental Fig. S11C), which may evidence differential IGHMBP2-dependent GCN2 activity or translation buffering effects that cannot be directly measured via net changes with this experiment.”

We then reconcile these results with our interpretation of pro-survival ISR in the 3rd paragraph of the Discussion:

“We surmise pro-survival ISR may enable cells to buffer upstream molecular effects of IGHMBP2 disruption. We were therefore surprised that up to 9 days of ISR suppression via ISRIB or GCN2iB treatment did not measurably affect proliferation rates of IGHMBP2-deficient cells (Supplemental Fig. S11A). We thus speculate that while the ISR alters homeostasis and translation in IGHMBP2 deletion cells—which is supported by morphological and translation rescue upon ISR suppression (Fig. 5C-D)—consequences of IGHMBP2 loss orthogonal to ISR stimulation may more directly influence proliferation rate.”

We also note here that testing 24 hr rather than 48 hr drug exchanges made no difference to results. Lastly, we wish to address that the HET clone in this experiment robustly showed a more sizable decline in growth rate relative to full deletion clones across triplicate wells among 3 different plates. We therefore removed the representative heterozygous trend in Fig. 1E to avoid misrepresenting its more consistent trends upon this further repetition. We note that translation reporter differences (Supplemental Fig. S11C) in HET cells reproducibly remain intermediately altered or unchanged as represented in Fig. 2E-F, while morphological differences appear stochastic and median cell size is enlarged at measurements after Day 3. (We note these cells were measured 48 hours after passaging, rather than 24 hours shown in Fig. 1D). While the more pronounced decline of HET cells occurs for unknown reasons, this does not change our overall conclusions (summarized below). The growth defect in HET Clone #2 may be a clone-specific feature—its generally enlarged morphology over time may be driven by relatively pronounced IGHMBP2-dependent cell cycle dysfunction for this clone.

We addressed this difference later in the 3rd paragraph of the Discussion:

“[...] consequences of IGHMBP2 loss orthogonal to ISR stimulation may more directly influence proliferation rate. IGHMBP2 HET cells also exhibited stochastic morphological differences and pronounced decline relative to KO cell growth, yet translation was largely unchanged over time relative to parental cells (Fig. 2E-F; Supplemental Fig. S11A-C). This demonstrates how growth processes may be particularly sensitive to deviations in IGHMBP2 abundance, while chronically defective translation depends on full loss of IGHMBP2.”

Reviewer #2

This manuscript describes the translational effects of deleting IGHMBP2, a gene linked with rare human diseases. Homozygous knockout cells grow slowly and show a modest reduction in overall protein synthesis. Specific mRNAs show larger translational changes, including genes encoding the translational machinery. Transcript-specific analysis also highlights chronic induction of ATF4, the transcription factor responsible for the ISR. This effect is substantially reversed when the ISR kinase Gcn2 is inhibited, although phosphorylation of its target, eIF2 α , is not detectably changed. Likewise, defects are not apparent in either tRNA levels or in altered ribosome profiles.

Taken together, these results strengthen the link between IGHMBP2 and translation, and point towards more specific hypotheses to explain the specific disease phenotypes caused by loss of IGHMBP2. The underlying data are clear, and support the conclusions even when the magnitude of effects is small. I think the readership of LSA will appreciate the contributions of this work and would support its publication, provided the concerns below can be addressed.

We sincerely thank Reviewer #2 for their positive assessment of our work and helpful feedback.

1. *The CellTiter-Glo assay in Figure S2A, appears to show growth defects in one heterozygous knockout (118) and one homozygous knockout (115), with much less effect in another heterozygous knockout (33) and another homozygous knockout (61). This seems to differ from the other proliferation assays, and these specific results are not discussed in the paper. Is there an explanation for this difference?*

We thank Reviewer #2 for the opportunity to elaborate on Fig. S2A. We originally set out to use the CellTiter-Glo assay to measure relative growth rates between cells differentially expressing IGHMBP2, but were unable to achieve consistent results with this assay despite consistent seeding between cell lines amid maintaining optimal passaging conditions. We considered this may be due to a number of reasons. Primarily, the CellTiter-Glo luminescence readout corresponds to ATP abundance and is thus foremostly a measure of the metabolic state as a function of cellular viability. (We have therefore revised the terminology in the S2A figure legend from “growth rate” to “metabolic” profile.) We suspected that CellTiter-Glo may not enable sensitive measurement of nuanced growth-rate changes that may require a longer timeframe to detect. This prompted us to re-strategize toward pursuing the competition assay in Fig. 1E, which would allow us to track relative growth rate across multiple cell passages. The growth competition results in Fig. 1E then confirmed that observable differences in relative fitness between cell lines requires measuring >1 week of timepoints.

We therefore show metabolic assay results in Fig. S2A only to support that we appropriately harvested cells during logarithmic growth (metabolic) phases of cell populations for OPP readings (Fig. 1D-E; S2B-C), as the OPP assay is sensitive to translational suppression that occurs once cells approach confluency. We did not further explore the evident S2A metabolic inconsistencies Reviewer #2 alludes to, although we suspect this may ultimately be explained by noisy IGHMBP2-dependent metabolic gene expression changes detected in our functional sequencing analyses. Below we have attached more example CellTiter-Glo time-courses (corresponding to samples collected for Figs. 2D-E) to demonstrate the metabolic noise between cell lines during exponential growth, which we note is atypical, as CellTiter-Glo otherwise yields very consistent results

between cell lines with more minor genetic perturbations in our experience. Further interrogation of IGHMBP2-dependent metabolic disruption may therefore be a future direction of interest.

(Figure S2A)

Additional CellTiter-Glo results

Related to this comment, additional experimentation in response to Reviewer #1-Comment #5 showed that the growth rate of HET Clone #2 is actually lesser than KO clones, and we therefore sought to address the variable proliferation effect size in the Discussion (paragraph 3).

2. If the flow cytometry data in 2D is plotted on a logarithmic scale, this should be made clear. And, if these data are indeed log-scaled, are the relative levels in 2E and 2F normalized as linear intensities? If these are relative values on a logarithmic scale (as in, the ratio of two log-scaled measurements) then some justification should be presented for this calculation.

We agree on the importance of clarifying flow cytometry scaling and have addressed Reviewer #2's labeling suggestions in relevant figures. The flow cytometry data in this study are scaled via *arcsinh* transformation prior to determining additional standardizations or normalizations (GFP/RFP, AlexaFluor 647(KO:WT), etc.). For Figs. 2E-F specifically, relative levels of fluorescence intensities are determined against reference samples per each day of measurement. We did not use data collected from different days or experiments to perform direct normalizations between data. Therefore, we expect the ratio between two log-scaled measurements to reflect experimental rather than instrumental, day-to-day signal variability. Furthermore, we only utilize scaled median fluorescence intensity values for subsequent quantifications, which are less affected by skew and outliers and generally accepted to proportionally reflect fluorescence fold-changes as opposed to using untransformed values in flow cytometry analysis.

We have adjusted the *Flow cytometry* methods section to better clarify this:

"Data values were arcsinh transformed and gated to remove debris and doublets. Derivation of standardized and normalized values were performed among data collected on the same day per each experiment."

3. Figure 3A shows a number of genes with large (mostly RNA level) changes. How much do the affected genes overlap between different clones? Looking at the data tables, it would seem that there is substantial overlap.

We thank Reviewer #2 for their interest that we expand our assessment of DEG correlation between clones. We have added a new Fig. S7A-B to demonstrate correlation among significant RNA-seq hits between clones. We added to the results:

"To visualize genes with reproducible RNA-seq level changes, we filtered for significant RNA-seq DEGs in both heterozygous or full IGHMBP2 deletion clones (Supplemental Fig. S7A-B). For each set, the majority of hits overlapped in the same directionality (10 out of 13 genes between IGHMBP2 HET clones; 43 out of 51 genes between IGHMBP2 KO clones) and showed significant correlation (Supplemental Fig. S7A-B). Together, these data identify the genetic changes underlying morphological and translation defects observed as a consequence of IGHMBP2 deletion, again implicating translation as a process impacted by loss of IGHMBP2 (Fig. 3E)."

We note that all genes with any significance categorization in Supplemental Table 1 were already filtered for significance to at least 2 of the 4 datasets for all categories; thus, to optimize discoverability of genes that a stronger threshold may discard, this filtering strategy is flexible to identifying hits significant in 1 HET + 1 KO rather than requiring significance in both HETs and KOs. As Reviewer #2 notes, substantial expression changes mostly appear at the RNA level, and we therefore do not perform overlap analysis for the Ribo-seq hits and instead refer to Fig. S8A to evaluate TE changes of interest determined upon conducting multivariate, Ribo:RNA-seq interaction analysis.

4. In Figure 4C, it would be helpful to display ATF4 CDS occupancy alongside the two uORFs. Also, because uORF2 overlaps the CDS, it would be helpful to specify whether the uORF 2 occupancy measurements included or excluded this overlapping region.

We agree with Reviewer #2 that showing CDS occupancy would enhance Fig. 4C and have updated this panel and methods. In our quantification, uORF2 occupancy measurements excludes the CDS overlapping region by omitting the 50 nt bin overlapping the CDS AUG (at chr22: 39,521,446) for both uORF2 and CDS. We set the uORF2 range to end at 39,521,400 and began the CDS range at 39,521,450. As Ribo-seq reads are less than 40 nt in length, excluding one bin would reliably ensure the same counts are not redundantly quantified. We also clarify here that the CPM normalization to visualize regional occupancy is distinct from, yet reproduced by the scaling methodology performed on raw counts during differential expression analysis.

We updated our methods:

“To determine localized read mapping coverage for specific genes, such as ATF4 uORFs and CDS [...] Normalized counts were extracted from .bed files per genomic regions analyzed corresponding to chr22: 39,520,650-39,520,700 (uORF1); 39,520,750-39,520,800 and 39,521,350-39,521,400 (uORF2, separated by splice-site); 39,521,450-39,522,600 (CDS).”

5. In Figure 5A and 5C, are all tests of significance made relative to "Parental" no-drug? Bars or brackets would be helpful to illustrate what comparison was performed. A two-way ANOVA might provide a better statistical analysis here.

We thank Reviewer #2 for their feedback - yes, t-testing was made relative to “Parental” no-drug for individual comparisons. For 5A and 5C (now 5A-B), we agree that two-way testing is of appropriate interest to address possible interaction effects between the treatment groups (DMSO, ISRIB, GCN2iB) compared to parental versus KO cells, especially as a significant interaction would be expected. Upon performing Shapiro-Wilk test and Breusch-Pagan test on this dataset, we find that our data do not follow normal distribution nor homoscedasticity required for parametric ANOVA. We therefore rank-transformed this data prior to running a two-way ANOVA. Results however were insignificant, and we therefore ultimately proceeded with one-way ANOVA (Kruskal-Wallis) on rank-transformed data to aid interpretation of the results, which we discuss:

“Although expected, two-way interaction between ISRIB and GCN2iB-treated parental versus IGHMBP2 deletion cells was statistically insignificant. This may suggest the effect of ISR inhibition on ATF4 reporter expression is not IGHMBP2-dependent, or is driven by the mild effect of ISRIB on parental cells and overall mild effect of GCN2iB on IGHMBP2 KO cells. One-way analysis, however, shows ATF4 upregulation in IGHMBP2 KO cell lines is partially rescued by GCN2iB treatment, supporting the involvement of GCN2 in ISR activation (Fig. 5A-B).”

We did utilize two-way ANOVA on new rank-transformed data added to the manuscript in Fig. 5C-D and Supplemental Fig. S11B, as results returned significant, and we could therefore proceed with Tukey post-hoc testing and use compact letter display to indicate pairwise relationships. We additionally thank Reviewer #2 for prompting us to reevaluate the use of t-testing among our other panels as well (Figs. 2E,F & Fig. 4F). We switched to one-

way ANOVA to handle analysis of multiple cell lines and catching that our median data was not parametric post-normalization with controls set to 1.

6. It is noted that "ribosomal machinery, translation initiation, and translation elongation factors" show significant translational change. This pattern is suggestive of changes in 5' TOP mRNA translation downstream of reduced mTOR activity, has this been considered?

We thank Reviewer #2 for prompting us to consider and explore mTOR activity in our work considering its connection to regulation of 5' TOP mRNAs associated with translation factor genes. For our response, we performed additional analysis and provide an extended discussion here:

To determine if we could detect evidence of change to mTORC1 activity, we cross-referenced mTORC1 targets in our Ribo:RNA-seq results via a custom GSEA using the mTORC1 signaling hallmark human gene set from the Molecular Signatures Database (Liberzon et al. 2014), which includes refined genes among multiple published mTORC1-associated DEGs.

Hallmark mTORC1 gene enrichment among IGHMBP2-dependent Ribo:RNA-seq (top panel) results was insignificant ($p_{\text{adj}} = 0.29$) with a normalized enrichment score of -1.21, and results were null when testing enrichment among RNA-seq results (bottom panel) under GSEA parameters described in the manuscript. The grey line indicates rank trends of the mTORC1 hallmark gene sets. As with our ATF4 target GSEA discussion, mTORC1-dependent genes may indeed be down-regulated in TE, yet beneath thresholds toward statistical confidence.

GSEA with hallmark mTORC1 gene set with IGHMBP2 KO TE (Ribo:RNA) results

GSEA with hallmark mTORC1 gene set with IGHMBP2 KO RNA-seq results

The 63 out of 200 hallmark mTORC1 genes from our TE GSEA are shown below. We list average Log2FoldChanges and corresponding p-adjusted values between both KO clones for the top 10 ranked genes. Among all 63 genes, only CD9 was found to be a significantly down-regulated DEG in RNA-seq (L2FC = -1.4, p.adj = 3.7E-6 and Ribo-seq (L2FC = -1.5, p.adj = 0.0027) results, yet in only KO clone#2. However, trends were consistent and statistics were relatively near cutoff thresholds in KO clone#1, as CD9 was down-regulated by L2FC = -0.35 in RNA-seq (p.adj = 0.026) and L2FC = -0.85 in Ribo-seq (p.adj = 0.13) results. This further exemplifies how statistical thresholds may lose possible DEG candidates in nuanced perturbation studies such as our work, yet we ultimately conclude there is weak evidence of IGHMBP2-dependent mTOR activity disruption from this cursory analysis.

GSEA results with hallmark mTORC1 set with IGHMBP2 KO TE (Ribo:RNA) results

Enrichment Rank	Gene	Log2FoldChange	p.adj	Enrichment Rank	Gene	Enrichment Rank	Gene
1	PSMG1	-0.125	0.99998	11	BCAT1	21	CCTGA
2	SSR1	-0.125	0.99998	12	HMGCR	22	ASNS
3	HSPD1	-0.127	0.99998	13	USO1	23	RAB1A
4	MTHFD2	-0.129	0.99998	14	HSPA9	24	PSMD12
5	RPA1	-0.131	0.99998	15	SERP1	25	PPA1
6	PSMB5	-0.134	0.99998	16	IDH1	26	PRDX1
7	NUP205	-0.137	0.62125	17	NFYC	27	UCHL5
8	CYB5B	-0.141	0.99998	18	PSMC6	28	NUFIP1
9	HPRT1	-0.141	0.99998	19	GBE1	29	PNP
10	COPS5	-0.143	0.99998	20	PITPNB	30	TFRC
Enrichment Rank	Gene	Enrichment Rank	Gene	Enrichment Rank	Gene	Enrichment Rank	Gene
31	ACTR3	41	GTF2H1	51	GCLC	61	ME1
32	ACTR2	42	PNO1	52	PLOD2	62	POLR3G
33	NAMPT	43	ETF1	53	ATP6V1D	63	AK4
34	DHFR	44	ADD3	54	UFM1		
35	EIF2S2	45	EEF1E1	55	IDH1		
36	CACYBP	46	SC5D	56	TBK1		
37	INSIG1	47	SEC11A	57	PSMA3		
38	IFRD1	48	PSMD14	58	HSPE1		
39	SKAP2	49	LTA4H	59	CTH		
40	PSMA4	50	HMGCS1	60	CD9		

Literature suggests mTOR and ISR signaling share ATF4-dependent overlap (Torrence et al. 2021), and thus decoupling mTORC1 versus ISR-dependent DEGs is challenging. Moreso, GCN2 activation has been shown to converge with mTORC1 signaling to up-regulate 5' TOP mRNAs (Damgaard & Lykke-Andersen, 2011). Considering IGHMBP2-dependent differential growth phenotypes, we would agree it seems feasible that growth disruption likely leverages growth signaling pathways such as mTOR to perturb 5' TOP mRNA expression. We also invoke a very relevant discussion in a key study by *Sidrauski et al.* (2015), where authors also observed that expression of abundant ribosomal proteins and elongation factors were suppressed upon ISR induction. *Sidrauski et al.* proposed that while these genes correspond to 5' TOP mRNAs that may be regulated by mTOR disruption, this would arguably occur downstream of eIF2 α phosphorylation in an ISR context where ISRIB treatment reverses effects. *Sidrauski et al.* also cited an intriguing alternative model where ISR-induced stress granules also regulate 5' TOP mRNAs in a GCN2 and mTOR-dependent mechanism.

We therefore added a note in our manuscript to better clarify the association between ribosomal proteins and elongation factor suppression with ISR activation:

“IGHMBP2-dependent, mild suppression of ribosomal proteins and elongation factors shown in our sequencing results would also be consistent as a repercussion of ISR-induced eIF2 α phosphorylation (Sidrauski et al. 2015).”

Our ISRIB data provides direct evidence of IGHMBP2-dependent ISR activation, and the involvement of modulated mTORC1 activity may be a direction of interest for further study. Without stronger evidence of mTOR dysregulation, we find mTOR discussion

ultimately beyond the scope of our current work, and thus we limit this discussion to our responses here.

As an aside on this topic, in addition to ATF4, we also initially noticed CNBP as a repressed Δ TE hit of interest (Fig. 4A). Intriguingly, when incorrectly expressed, CNBP causes myotonic dystrophy, suppresses 5' TOP mRNAs, and decreases global translation (Huichalaf et al. 2009). Given the similarity in associated diseases among disrupted CNBP and IGHMBP2, we asked very simply whether CNBP suppression was sufficient to activate ATF4 upregulation as a possible disease mechanism of IGHMBP2 by expressing two different guide RNAs targeting *CNBP* and leveraging the CRISPRi machinery in our parental ATF4 reporter cell line to repress CNBP expression. Our preliminary findings showed ATF4 was not upregulated in CNBP-silenced cells, and we therefore concluded that IGHMBP2-dependent ISR activation is orthogonal to CNBP suppression. Further experimental validation is needed to bolster the connection between IGHMBP2 and CNBP regulation and determine whether IGHMBP2-dependent CNBP downregulation plays a role in the 5' TOP mRNA suppression. Ultimately, these insights reinforce broad centering of translational dysregulation among various genetic diseases—whether by the interference of important translation factors, ISR activation, mTOR dysregulation, or plausibly multiple of these mechanisms occurring in combination.

7. It is proposed that eIF2 α phosphorylation is not elevated because of homeostatic regulation. Could effective homeostatic regulation also explain why codon-level effects are not seen in ribosome profiling—that is, translation is reduced to the extent needed to resolve these stalls?

We thank Reviewer #2 for raising this interesting possibility. As it is widely accepted that the ISR enables restoration of cellular homeostasis by attenuating translation (stress granules are proposed to be one example ISR mechanism in response to acute stress), it would therefore be relevant to emphasize this possibility considering our challenges in overcoming small effect sizes in our study. We have added this point as an additional possibility in 3rd paragraph of the Discussion:

"[...] thus measuring ISR activation in different cell types or animal models would be important future work. We note that acutely activated ISR promotes restoration of homeostasis through various proposed mechanisms, and it is interesting to postulate how chronic ISR-mediated translational suppression may function to promote survival by attenuating stressors such as eIF2 α phosphorylation and ribosomal stalling."

Minor points:

8. Some of the points on Figure 4C seem to overlap, is it possible to offset or jitter them slightly?

9. In Figure S3, should the legend in F read "Ribo-seq" rather than RNA-seq? This would better match the figure title, and explain how A-E differ from F-J

10. In the methods, it is written "tRNA-Tyr, tRNA-Val, or tRNA-Val". This would be less confusing if the anticodons were specified, so it was clear that two different isoacceptors were measured, rather than the repetition reflecting a typographical error.

We sincerely thank Reviewer #2 for their thoroughness in their minor points 8, 9, and 10, where they helpfully pointed out typos and visual adjustments which we have corrected.

April 26, 2024

RE: Life Science Alliance Manuscript #LSA-2023-02554-TR

Prof. Stephen N Floor
University of California, San Francisco
513 Parnassus Ave
San Francisco 94143

Dear Dr. Floor,

Thank you for submitting your revised manuscript entitled "IGHMBP2 deletion suppresses translation and activates the integrated stress response". We would be happy to publish your paper in Life Science Alliance pending final revisions necessary to meet our formatting guidelines.

- please be sure that the authorship listing and order is correct
- please add ORCID ID for the corresponding author -- you should have received instructions on how to do so
- please add callouts for all panels in Figures S2-S6 in the manuscript text

FIGURE CHECKS:

- please add sizes next to the blots in Figures 1 and 2

A. FINAL FILES:

B. MANUSCRIPT ORGANIZATION AND FORMATTING:

Thank you for your attention to these final processing requirements. Please revise and format the manuscript and upload materials within 4 days.

Sincerely,

May 2, 2024

RE: Life Science Alliance Manuscript #LSA-2023-02554-TRR

Prof. Stephen N Floor
University of California, San Francisco
513 Parnassus Ave
San Francisco 94143

Dear Dr. Floor,

Thank you for submitting your Research Article entitled "IGHMBP2 deletion suppresses translation and activates the integrated stress response". It is a pleasure to let you know that your manuscript is now accepted for publication in Life Science Alliance. Congratulations on this interesting work.

DISTRIBUTION OF MATERIALS:

Again, congratulations on a very nice paper. I hope you found the review process to be constructive and are pleased with how the manuscript was handled editorially. We look forward to future exciting submissions from your lab.

Sincerely,
